# Two Birds, One Stone: An Equivalent Transformation for Hyper-relational Knowledge Graph Modeling

## Abstract

By representing knowledge in a primary triple associated with additional attribute-value qualifiers, hyper-relational knowledge graph (HKG) that generalizes triple-based knowledge graph (KG) has been attracting research attention recently. Compared with KG, HKG is enriched with the semantic difference between the primary triple and additional qualifiers as well as the structural connection between entities in hyper-relational graph structure. However, to model HKG, existing studies mainly focus on either semantic information or structural information therein, fail to capture both simultaneously. To tackle this issue, in this paper, we propose an equivalent transformation for HKG modeling, referred to as TransEQ. Specifically, the equivalent transformation transforms a HKG to a KG, which considers both semantic and structural characteristics. Then a generalized encoder-decoder framework is developed to bridge the modeling research between KG and HKG. In the encoder part, KG-based graph neural networks are leveraged for structural modeling; while in the decoder part, various HKG-based scoring functions are exploited for semantic modeling. Especially, we design the sharing embedding mechanism in the encoder-decoder framework with semantic relatedness captured. We further theoretically prove that TransEQ preserves complete information in the equivalent transformation, and also achieves full expressivity. Finally, extensive experiments on three benchmarks demonstrate the superior performance of TransEQ in terms of both effectiveness and efficiency. On the largest benchmark WikiPeople, TransEQ significantly improves the state-of-the-art models by 15% on MRR.

## 1 Introduction

In the past decade, knowledge graph (KG) has been widely studied in artificial intelligence area (Ji et al., 2021). By representing facts into a triple of $(s, r, o)$ with subject entity $s$, object entity $o$ and relation $r$, KG stores real-world knowledge in a graph structure. However, recent studies find that KG with simple triples provides incomplete information (Galkin et al., 2020; Rosso et al., 2020). For example, both (*Alan Turing*, `educated at`, *Cambridge*) and (*Alan Turing*, `educated at`, *Princeton*) are true facts in KG, which might be ambiguous when the degree matters.

Hence, the hyper-relational KG (HKG) (Galkin et al., 2020; Rosso et al., 2020; Yu & Yang, 2021), a.k.a., knowledge hypergraph (Fatemi et al., 2020; 2021) and n-ary knowledge base (Guan et al., 2019; Liu et al., 2021), is proposed for more generalized knowledge representation. Formally, in HKG, a primary triple is augmented with additional attribute-value qualifiers for rich semantics, called the hyper-relational fact (Guan et al., 2020). Note that the triple without qualifiers is a special case of hyper-relational facts. Taking Figure 1 as an example, both (*Alan Turing*, `educated at`, *Cambridge*, (`degree`, *Bachelor*)) and (*Alan Turing*, `educated at`, *Princeton*, (`degree`, *PhD*)) are hyper-relational facts, where (`degree`, *Bachelor*) and (`degree`, *PhD*) are qualifiers with the degree attribute considered. Such hyper-relational facts are ubiquitous that over 1/3 of the entities in Freebase (Bollacker et al., 2008) involve in them (Wen et al., 2016).

To learn from HKG and further benefit the downstream tasks, HKG modeling learns low-dimensional vector representations (embeddings) of entities and relations (Wang et al., 2021), which designs a scoring function (SF) based on the embeddings to measure the hyper-relational fact plausibility such

that valid ones obtain higher scores than invalid ones. Especially, existing studies mainly consider two aspects of semantic information and structural information in HKG for modeling.

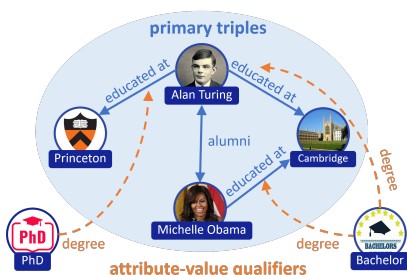

Figure 1: An example of a HKG including primary triples and attribute-value qualifiers. The entities/relation in the triple are called as primary entities/relation, and attributes/values in qualifiers are called as qualifier entities/relations.

The semantic information emphasizes the interaction between entities and relations in a hyper-relational fact. Especially, there is a distinction, a.k.a., semantic difference (Galkin et al., 2020) between the primary triple and attribute-value qualifiers, e.g., the primary triple (*Alan Turing*, `educated at`, *Cambridge*) serves as the fundamental part and preserves the essential knowledge of Alan Turing's education experience at Cambridge, while the attribute-value qualifier (`degree`, *Bachelor*) serves as the auxiliary part and enriches the primary triple. To model the semantic information, early studies treat the primary relation and qualifier relations as an n-ary (n≥2) composed relation (Abboud et al., 2020) or multiple semantically equal attributes (Guan et al., 2019; Liu et al., 2021), largely ignoring the semantic difference. Various SFs are further developed in recent studies (Galkin et al., 2020; Rosso et al., 2020; Yu & Yang, 2021) with semantic difference considered.

On the other hand, the structural information focuses on the topological connection between entities in the hyper-relational graph structure, like an entity's neighboring entities under various hyper-relational links, e.g., in Figure 1 *Bachelor* and *Michelle Obama* are neighbors of *Alan Turing* via `degree` and `alumni`, respectively. Only few studies (Galkin et al., 2020; Yadati, 2020) extend hypergraph neural network (HGNN) based modules to capture the structural information in HKG, however, empirical results in (Yu & Yang, 2021) demonstrate that removing such modules will not bring performance degradation, i.e., the direct extensions are quite immature for effective structural information capture. Hence, to the best of our knowledge, none of existing studies achieve HKG modeling with both semantic information and structural information completely captured, and it is still an open problem to be addressed.

Targeting on this open problem, we look back to KG modeling with an interesting observation that, recent studies (Vashishth et al., 2019; Yu et al., 2021) leverage an encoder-decoder framework for KG modeling, i.e., a powerful graph neural network (GNN) based encoder and an expressive SF-based decoder on triples are leveraged for structural information and semantic information, respectively. Inspired by this, in this paper, we propose an EQuivalent Transformation for HKG modeling, termed as TransEQ. Specifically, TransEQ designs an equivalent transformation on the hyper-relational graph structure, transforming a HKG to a KG with semantic difference considered, based on which a generalized encoder-decoder framework is further developed to capture information. For structural information, TransEQ introduces a GNN-based encoder on transformed KG with transformation characteristics combined. As for semantic information, to measure the plausibility of a hyper-relational fact, TransEQ exploits various SFs in existing HKG modeling studies as the decoder. The sharing embedding mechanism is further designed to capture the semantic relatedness between hyper-relational facts. In this way, with the equivalent transformation, the encoder-decoder framework in TransEQ captures not only structural information but also semantic information, which is the very innovation of this work, just like killing two birds with one stone. Besides, the flexible choice of SF in decoder ensures the full expressivity of TransEQ, representing all types of relations. We further theoretically prove that the proposed transformation is equivalent between a HKG and a KG without information loss. Extensive experiments show that TransEQ achieves the state-of-the-art results, obtaining a 15% relative increase of MRR on the largest benchmark WikiPeople.

## 2 RELATED WORK

As described before, related studies mainly exploit two aspects of semantic information and structural information for HKG modeling, considering HKG-based SF design and hyper-relational graph structure, respectively.

**Semantic Modeling Studies.** Given a hyper-relational fact (*Alan Turing*, `educated at`, *Cambridge*, (`degree`, *Bachelor*)), some studies treat all involved relations as an n-ary composed relation `educated at_degree` (here n is 3) with the fact (`educated at_degree`, *Alan Turing*, *Cambridge*, *Bachelor*). For example, both m-TransH (Wen et al., 2016) and RAE (Zhang et al., 2018) extend the SF of TransH (Wang et al., 2014) to the hyper-relational case. BoxE (Abboud et al., 2020) combines translational idea with box embeddings. Moreover, GETD (Liu et al., 2020) and S2S (Di et al., 2021) are both generalized from TuckER (Balazevic et al., 2019), where GETD further introduces tensor ring decomposition while S2S applies neural architecture search techniques. The bilinear product is also extended to multilinear product with symmetric embeddings in m-DistMult (Yang et al., 2015), convolutional filters in HypE (Fatemi et al., 2020), and relational algebra operations in ReAlE (Fatemi et al., 2021). These studies are directly extended from KG modeling methods without multiple relational semantics considered. On the other hand, NaLP (Guan et al., 2019) and RAM (Liu et al., 2021) decompose all involved relations into semantically equal attributes, and treat the example fact into a collection of attribute-value qualifiers, (`educated at_head`, *Alan Turing*, `educated at_tail`, *Cambridge*, `degree`, *Bachelor*). Nevertheless, models above largely ignore the semantic difference in hyper-relational facts. To capture such semantic difference between the primary triple and attribute-value qualifiers, NeuInfer (Guan et al., 2020) and HINGE (Rosso et al., 2020) design two sub-modules for HKG modeling, i.e., one for triple modeling and the other one for qualifier modeling, where NeuInfer mainly adopts fully connected layers while HINGE resorts to convolutional neural networks. Besides, recent studies of GRAN (Wang et al., 2021) and Hy-Transformer (Yu & Yang, 2021) leverage transformer and embedding processing techniques for HKG modeling. However, these neural network based models rely on tremendous parameters for expressivity and are prone to overfitting.

**Structural Modeling Studies.** G-MPNN (Yadati, 2020) ignores attribute information and treats HKG as a multi-relational ordered hypergraph with n-ary composed relations, and further proposes multi-relational HGNN for modeling. The rough design makes G-MPNN less competitive in practice. StarE (Galkin et al., 2020) firstly introduces GNN for HKG modeling with a relation-specific message passing mechanism developed. However, StarE aggregates hyper-relational fact messages for a specific entity only when the entity involves with the primary triple, but ignores the ones when the entity is in attribute-value qualifiers, i.e., StarE only captures connections among primary triples (Yu & Yang, 2021). Thus, capturing structural information for HKG modeling is still immature and needs further investigation.

Overall, existing HKG modeling studies are affected by various limitations from semantics and structure, while our proposed TransEQ elegantly models both aspects with full expressivity achieved, which is a quite important property for learning capacity in both KG modeling (Balazevic et al., 2019; Sun et al., 2019) and HKG modeling (Abboud et al., 2020; Liu et al., 2020). Besides, the inductive link prediction and logical query for HKG are investigated in recent studies (Ali et al., 2021; Alivanistos et al., 2022), which are beyond the scope of this paper.

## 3 METHOD

Here we first introduce the mathematical definition of HKG as well as the investigated problem.

**Definition 1** *Hyper-relational Knowledge Graph. A HKG is defined as $\mathcal{G}^H = (\mathcal{E}, \mathcal{R}, \mathcal{F}^H)$, where $\mathcal{E}$ and $\mathcal{R}$ are the sets of entities and relations, respectively. A hyper-relational fact can be expressed as $(s, r, o, \{(a_i, v_i)\}_{i=1}^n)$, where $(s, r, o)$ is the primary triple and $\{(a_i, v_i) | a_i \in \mathcal{R}, v_i \in \mathcal{E}\}_{i=1}^n$ is the attribute-value qualifier set. Moreover, $\mathcal{F}^H \subseteq \mathcal{E} \times \mathcal{R} \times \mathcal{E} \times \mathcal{P}$ denotes the fact set and $\mathcal{P}$ denotes all possible combinations of attribute-value qualifiers.*

Note that the number of qualifiers can be zero for a hyper-relational fact, i.e., HKG reduces to KG with an empty set $\mathcal{P}$. In practice, attributes and values are also described by relations and entities, respectively (Galkin et al., 2020; Yu & Yang, 2021). Then we state our research problem.

**Problem 1** *HKG Modeling Problem. Given a HKG $\mathcal{G}^H = (\mathcal{E}, \mathcal{R}, \mathcal{F}^H)$, the HKG modeling problem aims to learn representations for entities and relations in $\mathcal{E}$ and $\mathcal{R}$, respectively.*

Especially, the HKG is always incomplete, which specifies the research problem as HKG completion problem in practice, i.e., given an incomplete hyper-relational fact with an entity missing at triple or

qualifiers, inferring the missing entity from $\mathcal{E}$ with observable facts $\mathcal{F}^H$. Moreover, HKG involves semantic information of primary triple and attribute-value qualifiers as well as structural information of hyper-relational graph structure, which should be elegantly considered in modeling.

As described before, the encoder-decoder framework has shown superior performance to capture both structural information and semantic information in KG (Yu et al., 2021; Vashishth et al., 2019), and thus a natural idea is to explore it for HKG modeling. Besides, standard RDF reification in semantic web (Frey et al., 2019) as well as compound value type in Freebase (Bollacker et al., 2008) are investigated to describe triple with metadata by transformation. These works provide a motivation to our work transforming a HKG to a KG with the encoder-decoder framework combined. Hence, we build TransEQ with such points in mind, which is presented in the following.

### 3.1 THE TRANSEQ MODEL

We now come to the details of TransEQ, the architecture of which is illustrated in Figure 2. TransEQ first introduces the equivalent transformation with a HKG transformed to a KG, and then develops a generalized encoder-decoder framework, where a GNN-based encoder and a SF-based decoder are leveraged for modeling structural information and semantic information, respectively.

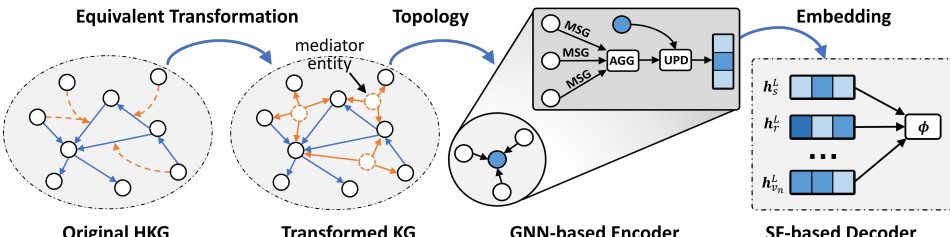

Figure 2: The architecture of our proposed HKG modeling model TransEQ.

### 3.1.1 ONE STONE: EQUIVALENT TRANSFORMATION

To identify the importance of transformation between HKG and KG, here we first introduce the definition of equivalent transformation.

**Definition 2** *Equivalent Transformation. A transformation between HKG and KG is equivalent, if the transformation preserves the complete information, i.e., given any HKG and its transformed KG via the transformation, they can be retrieved from each other.*

Moreover, a hyper-relational fact $(s, r, o, \{(a_i, v_i)\}_{i=1}^n)$ can be viewed as a hyper-relational edge, which connects entities of $s, o, \{v_i\}_{i=1}^n$ with heterogeneous semantics of primary relation $r$ and attributes $\{a_i\}_{i=1}^n$, as shown in Figure 3(a) with the $k$-th fact in a HKG. Thus, motivated by star expansion, we propose an equivalent transformation for hyper-relational edges such that entities and relations in the original HKG are reorganized to the transformed KG with both structural information and semantic information preserved.

Specifically, the equivalent transformation in Figure 3(b) introduces a mediator entity $b_k$ to identify the fact, and the primary relation $r$ is extended with two relations $r^{\text{sub}}$ and $r^{\text{obj}}$ for the relational edges between $b_k$ and subject entity $s$ and object entity $o$, respectively. The attribute information in original hyper-relational fact is preserved by the attributed-based edges between $b_k$ and value entities. Moreover, a relational edge $r$ connects entities $s$ and $o$ for semantic difference, i.e., such operation leads to a three-node clique motif (Milo et al., 2002), reflecting the primary role of the triple.

For better understanding, we present the execution process of the equivalent transformation in Algorithm 1. Especially, in lines 7-8, TransEQ utilizes different transformation operations to model the semantic difference. Besides, the original structure of the triple fact, i.e., hyper-relational fact without qualifiers, is kept to avoid redundancy. As proved later, such transformation brings no information loss, and provides a good basis for the following encoder-decoder framework.

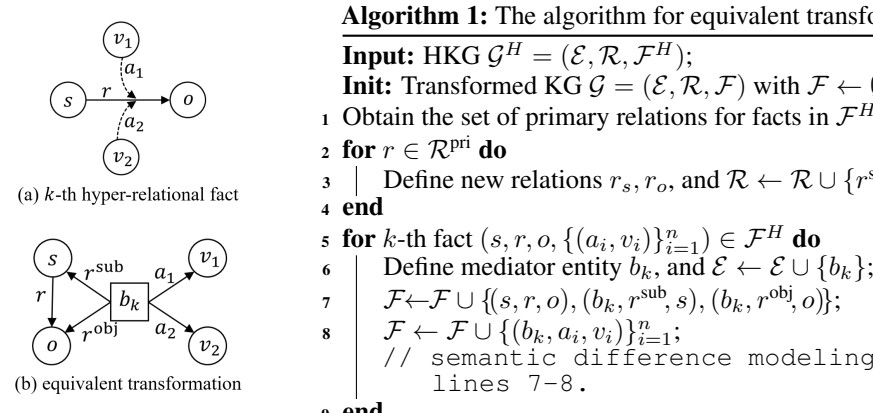

(a) $k$-th hyper-relational fact

(b) equivalent transformation

Figure 3: The illustration of the equivalent transformation.

**Algorithm 1:** The algorithm for equivalent transformation.

**Input:** HKG $\mathcal{G}^H = (\mathcal{E}, \mathcal{R}, \mathcal{F}^H)$;
**Init:** Transformed KG $\mathcal{G} = (\mathcal{E}, \mathcal{R}, \mathcal{F})$ with $\mathcal{F} \leftarrow \emptyset$;

1 Obtain the set of primary relations for facts in $\mathcal{F}^H$, $\mathcal{R}^{\mathrm{pri}}$;
2 **for** $r \in \mathcal{R}^{\mathrm{pri}}$ **do**
3     Define new relations $r_s, r_o$, and $\mathcal{R} \leftarrow \mathcal{R} \cup \{r^{\mathrm{sub}}, r^{\mathrm{obj}}\}$;
4 **end**
5 **for** $k$-th fact $(s, r, o, \{(a_i, v_i)\}_{i=1}^n) \in \mathcal{F}^H$ **do**
6     Define mediator entity $b_k$, and $\mathcal{E} \leftarrow \mathcal{E} \cup \{b_k\}$;
7     $\mathcal{F} \leftarrow \mathcal{F} \cup \{(s, r, o), (b_k, r^{\mathrm{sub}}, s), (b_k, r^{\mathrm{obj}}, o)\}$;
8     $\mathcal{F} \leftarrow \mathcal{F} \cup \{(b_k, a_i, v_i)\}_{i=1}^n$;
    `// semantic difference modeling in`
      `lines 7-8.`
9 **end**
**Output:** Transformed KG $\mathcal{G} = (\mathcal{E}, \mathcal{R}, \mathcal{F})$.

### 3.1.2 TWO BIRDS: ENCODER-DECODER FRAMEWORK

To model both structural information and semantic information in the original HKG, TransEQ further introduces a generalized encoder-decoder framework on the transformed KG.

**GNN-based Encoder.** The powerful GNN is developed to capture structural information, where the semantic relatedness in HKG and the mediator entities in equivalent transformation are also incorporated therein. Especially, the semantic relatedness explicitly lies in the shared primary relations across hyper-relational facts. For example, the hyper-relational facts of (*Alan Turing*, `educated at`, *Cambridge*, (`degree`, *Bachelor*)) and (*Alan Turing*, `educated at`, *Princeton*, (`degree`, *PhD*)) share the same primary relation, which indicates a strong semantic relatedness. On the other hand, in our proposed equivalent transformation, each mediator entity plays an important role of relaying connections among the entities in an original hyper-relational fact, and thus mediator entities aggregate the semantics of corresponding facts.

Hence, we introduce the sharing embedding for mediator entities to capture the semantic relatedness, and further combine it with three steps of unified multi-relational message passing mechanism in KG-based GNN (Schlichtkrull et al., 2018; Vashishth et al., 2019).

- *Initialized embedding.* Given the embedding dimension $d$, for the mediator entity $b$, we denote $\psi(b)$ the mapping from $b$ to its involved primary relation, and initialize its representation as $h_b^0 = [e_{\psi(b)}; e_b]$, where $e_{\psi(b)} \in \mathbb{R}^{\lfloor \alpha \cdot d \rfloor}$ and $e_b \in \mathbb{R}^{d - \lfloor \alpha \cdot d \rfloor}$ are sharing and independent embeddings, respectively. $\alpha$ is the hyperparameter to tune the sharing embedding ratio. Thus, mediator entities involved with the same primary relation $\psi(b)$ share part of embedding $e_{\psi(b)}$.

- *Message calculation.* Considering the stacking layers of GNN, we denote $m_{urt}^{l+1,\mathrm{ent}}$ and $m_{urt}^{l+1,\mathrm{rel}}$ the messages from a triple $(u, r, t)$ for target entity $t$ and relation $r$ at the $(l+1)$-th layer, respectively, which are calculated as follows,

$$m_{urt}^{l+1,\mathrm{ent}} = \mathrm{MSG}^{\mathrm{ent}}(h_u^l, h_r^l, h_t^l), \quad m_{urt}^{l+1,\mathrm{rel}} = \mathrm{MSG}^{\mathrm{rel}}(h_u^l, h_r^l, h_t^l),$$

where $h_u^l, h_t^l, h_r^l \in \mathbb{R}^d$ are the embeddings of entities and relation at the $l$-th layer, while $\mathrm{MSG}^{\mathrm{ent}}$ and $\mathrm{MSG}^{\mathrm{rel}}$ can be composition function in CompGCN (Vashishth et al., 2019), relation-specific projection in R-GCN (Schlichtkrull et al., 2018) and etc. Besides, the entity representations at the input layer are expressed as,

$$h_x^0 = \begin{cases} [e_{\psi(x)}; e_x] & \text{if } x \text{ is mediator entity} \\ e_x' \in \mathbb{R}^d & \text{if } x \text{ is original entity} \end{cases}, \text{ for } x \in \{u, t\},$$

- *Message aggregation.* Then neighborhood messages of $M_t^{l+1}$ and $M_r^{l+1}$ are aggregated as follows,

$$M_t^{l+1} = \mathrm{AGG}^{\mathrm{ent}}(m_{urt}^{l+1,\mathrm{ent}} \mid r \in \mathcal{R}, u \in \mathcal{N}_t^r), \quad M_r^{l+1} = \mathrm{AGG}^{\mathrm{rel}}(m_{urt}^{l+1,\mathrm{rel}} \mid (u, t) \in \mathcal{N}_r),$$

where $\mathcal{N}_t^r$ denotes the entities linked to $t$ via relation $r$ and $\mathcal{N}_r$ denotes the entity pair linked by relation $r$. $\mathrm{AGG}^{\mathrm{ent}}$ and $\mathrm{AGG}^{\mathrm{rel}}$ are aggregation functions like mean/sum pooling function.

- *Representation update.* Finally, the representations at the $(l + 1)$-th layer are updated with aggregated messages and former layer representations:

$$\boldsymbol{h}_t^{l+1} = \text{UPD}^{\text{ent}}(\boldsymbol{M}_t^{l+1}, \boldsymbol{h}_t^l), \; \boldsymbol{h}_r^{l+1} = \text{UPD}^{\text{rel}}(\boldsymbol{M}_r^{l+1}, \boldsymbol{h}_r^l),$$

where $\text{UPD}^{\text{ent}}$ and $\text{UPD}^{\text{rel}}$ can be nonlinear activation functions.

Owing to above encoding process, TransEQ fully exploits the topological connections between entities for structural information.

**SF-based Decoder.** The decoder part exploits various SFs to model semantic information. For each hyper-relational fact, the encoder part feeds the representations of corresponding entities and relations into the SF-based decoder to model the interaction between entities and relations therein. Especially, the choice of SF is orthogonal to the encoder, and most existing SFs on HKG modeling can be modified in the decoder. For example of the hyper-relational fact $x := (s, r, o, \{(a_i, v_i)\}_{i=1}^n) \in \mathcal{F}^H$, we rewrite m-DistMult's SF as:

$$\phi(x) = \langle \varphi(\boldsymbol{h}_r^L, \boldsymbol{h}_{a_1}^L, \cdots, \boldsymbol{h}_{a_n}^L), \boldsymbol{h}_s^L, \boldsymbol{h}_o^L, \boldsymbol{h}_{v_1}^L, \cdots, \boldsymbol{h}_{v_n}^L \rangle,$$

where $\phi(x)$ is plausibility score measured by TransEQ, $\langle \cdot \rangle$ denotes the multilinear product[1], and $L$ denotes the number of GNN layers in encoder part. Since m-DistMult adopts a composed relation for SF, we introduce the function $\varphi$ to aggregate the embeddings of involved primary relation and attributes for the composed relation embedding, such as mean/sum pooling function. Note that the semantic difference in HKG is also modeled by the SF in decoder. Various SFs are further investigated in experiments later.

**Model Training.** To learn the model parameters, we adopt the cross-entropy loss for training. For the hyper-relational fact $x \in \mathcal{F}^H$ with $\phi(x)$, the practical loss can be written as:

$$\mathcal{L} = \sum_{x \in \mathcal{F}^H} \mathcal{L}_x(\phi) = \sum_{x \in \mathcal{F}^H} - \log \frac{e^{\phi(x)}}{e^{\phi(x)} + \sum_{x' \in \mathcal{N}_x} e^{\phi(x')}}, \tag{1}$$

where $\mathcal{N}_x$ denotes the negative samples, i.e., entities in triple and attribute-value qualifiers of $x$ are replaced by other entities in $\mathcal{E}$. The training algorithm of TransEQ is presented in Appendix B for better understanding. The overall model is trained in a mini-batch way with batch normalization and dropout utilized for regularization.

Overall, the proposed TransEQ develops an equivalent transformation that transforms a HKG to a KG. Then a generalized encoder-decoder framework associates KG modeling research with HKG ones, where KG-based GNN encodes structural information while HKG-based SF in decoder focuses on semantic information.

Table 1: A comparison of representative HKG modeling studies. $n_e$, $n_r$ and $n_r^{\text{pri}}$ denote the numbers of entities, relations and primary relations. $d$ is the embedding dimension. $n_a$ is the maximum number of attribute-value qualifiers for facts, and $N = |\mathcal{F}^H|$ is the total number of facts in HKG. Neural: neural network based SF, Multilinear: multilinear product based SF.

| Model | Structure Modeling | Semantic Difference | Scoring Function | Expressive | $\mathcal{O}_{\text{time}}$ | $\mathcal{O}_{\text{space}}$ |
|---|---|---|---|---|---|---|
| NaLP | ✗ | ✗ | Neural | ✗ | $\mathcal{O}(d^2)$ | $\mathcal{O}(n_e d + n_r d)$ |
| m-DistMult | ✗ | ✗ | Multilinear | ✗ | $\mathcal{O}(d)$ | $\mathcal{O}(n_e d + n_r^{\text{pri}} d)$ |
| HypE | ✗ | ✗ | Multilinear | ✓ | $\mathcal{O}(d)$ | $\mathcal{O}(n_e d + n_r^{\text{pri}} d)$ |
| HINGE | ✗ | ✓ | Neural | ✗ | $\mathcal{O}(d^2)$ | $\mathcal{O}(n_e d + n_r d)$ |
| G-MPNN | HGNN | ✗ | Multilinear | ✗ | $\mathcal{O}(Nd^2)$ | $\mathcal{O}(n_e d + n_r^{\text{pri}} d + n_a d)$ |
| StarE | GNN | ✓ | Neural | ✗ | $\mathcal{O}(Nd^2 + n_a d^2)$ | $\mathcal{O}(n_e d + n_r d)$ |
| TransEQ | Transformation & GNN | ✓ | Arbitrary SF | ✓ | $\mathcal{O}(Nd^2)$ | $\mathcal{O}(n_e d + n_r d + Nd)$ |

## 3.2 Theoretical Understanding

**Complexity Analysis.** To distinguish our proposed TransEQ model design, in Table 1, we present a comparison of HKG modeling studies with structural modeling, semantic modeling, full expressivity

---

[1] $\langle \boldsymbol{h}_1, \boldsymbol{h}_2, \cdots, \boldsymbol{h}_n \rangle = \sum_i \boldsymbol{h}_1[i] \, \boldsymbol{h}_2[i] \cdots \boldsymbol{h}_n[i]$

as well as time and space complexity. According to the table, structural information is rarely explored in existing studies, while HGNN in G-MPNN is at its early stage and thus fails to model attribute semantics. StarE only captures triple-based connections (Yu & Yang, 2021), while TransEQ combines the equivalent transformation with GNN for structural information. As for semantic modeling, TransEQ not only captures the semantic difference between the primary triple and attribute-value qualifiers, but also applies for arbitrary SFs. Compared with the weak expressive power of most studies, the flexible choice of SF guarantees the full expressivity of TransEQ to model various HKGs, and brings performance improvement. Besides, the message passing mechanism in modeling structural information leads to the time complexity of $\mathcal{O}(Nd^2)$, and Transformer module in StarE brings an additional complexity of $\mathcal{O}(n_a d^2)$. Since the equivalent transformation introduces a mediator entity for each hyper-relational fact, TransEQ builds the space complexity of $\mathcal{O}(n_e d + n_r d + N d)$ with original entities and relations considered. Owing to parallel implementation and GPU acceleration, TransEQ obtains comparable efficiency to the fastest current studies in experiments. In this way, TransEQ achieves efficient and expressive HKG modeling with both structural information and semantic information captured.

**Information Preserving Transformation.** Following the structural information loss concern in hyperedge expansion (Arya et al., 2021; Dong et al., 2020; Zhou et al., 2006), here we investigate the information loss problem for our proposed transformation on HKG, which emphasizes on preserving both structural information and semantic information. Based on the equivalent transformation in Definition 2 and the proposed transformation in TransEQ, we identify the property with following theorem.

**Theorem 1** *In the conversion from a HKG to a KG, the proposed transformation in TransEQ is an equivalent transformation and preserves the complete information.*

**Full Expressivity.** To demonstrate the expressivity of TransEQ, here we introduce the full expressivity property (Abboud et al., 2020; Fatemi et al., 2020; Liu et al., 2021). A HKG modeling model is fully expressive if, for any given HKG, the model can separate valid hyper-relational facts from invalid ones by appropriate parameter configuration. Considering the encoder-decoder framework in TransEQ, such property is mainly determined by the SF in decoder part, thus we establish the expressivity of TransEQ with the following theorem.

**Theorem 2** *With encoder parameters configured appropriately, the expressivity of TransEQ is in accord with that of the scoring function it uses in decoder, i.e., TransEQ is fully expressive if the scoring function used in decoder is fully expressive.*

Thus, with appropriate choice of SF like HypE (Fatemi et al., 2020) as well as model parameters, a fully expressive TransEQ model has the potential to represent all types of relations in HKG including symmetric relations, inverse relations, etc. (Liu et al., 2021; Sun et al., 2019), which generally outperforms the weak ones in practice, as validated in Section 4.3.

The proofs of above theorem are provided in Appendix C

**Manually-designed v.s. Learnable Transformations.** According to the TransEQ model design, with theoretical guarantee on preserving information, the manually-designed equivalent transformation paves the way for capturing both semantic information and structural information in HKG. Although such transformation design can be learnable, the learning process is over complex without theoretical guarantee, while TransEQ with the manually-designed transformation has achieved the state-of-the-art performance, as validated by results in Section 4.2. Moreover, the simple yet effective manually-designed transformation takes the semantic difference into consideration, which offers valuable insights and rethinking discussion to the HKG modeling research.

## 4 EXPERIMENTS AND RESULTS

### 4.1 EXPERIMENTAL SETUP

**Datasets.** The experiments are conducted on three benchmark HKG datasets, i.e., WikiPeople (Guan et al., 2019), JF17K (Zhang et al., 2018) and FB-AUTO (Fatemi et al., 2020). We follow the standard splits (Guan et al., 2019) of these datasets. Detailed statistics can be found in Appendix D.

**Baselines.** As for performance comparison, we compare with several state-of-the-art HKG modeling approaches, including semantic modeling ones of BoxE (Abboud et al., 2020) S2S (Di et al., 2021), HypE (Fatemi et al., 2020), NeuInfer (Guan et al., 2020), RAM (Liu et al., 2021), HINGE (Rosso et al., 2020), m-TransH (Wen et al., 2016) as well as structural modeling ones of StarE (Galkin et al., 2020) and G-MPNN (Yadati, 2020). Besides, in TransEQ, we mainly adopt CompGCN (Vashishth et al., 2019) as encoder and m-DistMult (Fatemi et al., 2020) as decoder.

**Task and Evaluation Metrics.** Following typical settings (Abboud et al., 2020; Fatemi et al., 2020; Guan et al., 2019; Liu et al., 2020; Wang et al., 2021), we evaluate HKG modeling approaches on HKG completion task in transductive setting, and predict the missing entity at each position including triple and qualifier parts. Note that this task is more generalized than only predicting positions in triple part (Galkin et al., 2020; Rosso et al., 2020; Yu & Yang, 2021). As for evaluation metrics, the standard mean reciprocal ranking (MRR) and Hit@1,3,10 are utilized in filtered setting (Bordes et al., 2013; Guan et al., 2019). Code and data available: `https://anonymous.4open.science/r/TransEQ_Implementation-03FB`.

## 4.2 HKG COMPLETION RESULTS

Table 2: Results of HKG completion on all datasets. Results of baselines are collected from original papers and (Di & Chen, 2022; Fatemi et al., 2020; Liu et al., 2021). Best results are highlighted in bold, and second best results are highlighted with underlines. "-" denotes missing results.

| Model | WikiPeople | | | JF17K | | | FB-AUTO | | |
|---|---|---|---|---|---|---|---|---|---|
| | MRR | Hit@1 | Hit@10 | MRR | Hit@1 | Hit@10 | MRR | Hit@1 | Hit@10 |
| m-TransH | - | - | | 0.444 | 0.370 | 0.581 | 0.727 | 0.728 | 0.728 |
| HINGE | 0.333 | 0.259 | 0.477 | 0.473 | 0.397 | 0.618 | 0.678 | 0.630 | 0.765 |
| NeuInfer | 0.350 | 0.282 | 0.467 | 0.517 | 0.436 | 0.675 | 0.737 | 0.700 | 0.805 |
| m-DistMult | 0.318 | 0.213 | 0.478 | 0.452 | 0.375 | 0.599 | 0.784 | 0.745 | 0.845 |
| HypE | 0.292 | 0.162 | 0.502 | 0.507 | 0.421 | 0.669 | 0.804 | 0.774 | 0.856 |
| RAM | 0.370 | 0.293 | 0.507 | 0.539 | 0.463 | 0.690 | 0.830 | 0.803 | 0.876 |
| S2S | 0.372 | 0.277 | 0.533 | 0.528 | 0.457 | 0.690 | - | - | - |
| BoxE | 0.395 | 0.293 | 0.503 | 0.560 | 0.472 | **0.722** | 0.844 | 0.814 | 0.898 |
| G-MPNN | 0.367 | 0.258 | 0.526 | 0.530 | 0.459 | 0.688 | 0.763 | 0.724 | 0.838 |
| StarE | 0.378 | 0.265 | 0.542 | 0.542 | 0.454 | 0.685 | 0.764 | 0.725 | 0.838 |
| TransEQ | **0.454** | **0.373** | **0.593** | **0.569** | **0.489** | **0.722** | **0.870** | **0.842** | **0.909** |

We present the benchmark comparison of HKG completion in Table 2. According to the results, our proposed TransEQ model achieves the state-of-the-art performance on all benchmarks. On the hardest dataset WikiPeople with the most entities and relations, TransEQ significantly improves the best baseline (BoxE) by 27% and 15% on Hit@1 and MRR, respectively. Considering hyper-relational connections provided in WikiPeople, this improvement demonstrates that our proposed equivalent transformation preserves complete HKG information. Besides, TransEQ significantly outperforms m-DistMult, its original decoder model without GNN-based encoder, which indicates the effectiveness and necessity to consider structural information in HKG modeling. Such results also imply that with powerful SFs like BoxE, TransEQ can obtain even better performance. Moreover, compared with structural modeling approaches of G-MPNN and StarE, the substantial improvement of TransEQ owes to subtle design of the equivalent transformation as well as the semantic information captured in the decoder part.

## 4.3 ENCODER-DECODER CHOICE COMPARISON

To further investigate the effects of different GNN-based encoders along with HKG-based SFs as decoders, we compare the performance of different encoder-decoder choices in Table 3. In the table, each result corresponds to the TransEQ model with **X** as encoder and **Y** as decoder.

According to the results of each row in Table 3, compared with original models (**X**=No Encoder), TransEQ models with various GNN-based encoders bring substantial improvement, which again demonstrates the effectiveness of structural information encoding. Since neural network models

Table 3: Performance comparison of different encoder-decoder choices on JF17K. "OOM" indicates out of memory. Best results for each encoder are highlighted in bold.

| | JF17K | | | | | |
|---|---|---|---|---|---|---|
| **Encoder X →** | CompGCN | | R-GCN | | No Encoder | |
| **Decoder Y ↓** | MRR | Hit@10 | MRR | Hit@10 | MRR | Hit@10 |
| m-TransH | 0.542 | 0.694 | OOM | | 0.444 | 0.581 |
| Transformer | 0.526 | 0.677 | OOM | | 0.504 | 0.648 |
| m-DistMult | 0.569 | **0.722** | 0.483 | 0.632 | 0.452 | 0.599 |
| HypE | **0.572** | 0.715 | **0.550** | **0.702** | **0.507** | **0.669** |

with tremendous parameters easily overfit, the performance improvement of GNN-based encoder for Transformer is much lower than that for other models. As for the encoder in each column, the decoder choices of HypE achieve the best performance, mainly attributed to the linear complexity and full expressivity property. Benefited from the generalized encoder-decoder framework, TransEQ can flexibly adapt to various GNNs and SFs for both superior performance and full expressivity.

### 4.4 INFORMATION SHARING STUDY

To validate whether the semantic relatedness in HKG is captured by sharing embedding on mediator entities, we obtain hyper-relational facts of top ten primary relations and visualize their mediator entity embeddings via t-SNE (Maaten & Hinton, 2008), as shown in Figure 4(a). We select WikiPeople for visualization considering explicit attribute information therein, and mediator entities belonging to the same primary relation are marked in the same color. From the figure, we observe that mediator entities are neatly clustered according to their mapping primary relations, which is in accord with our sharing embedding design in GNN-based encoder.

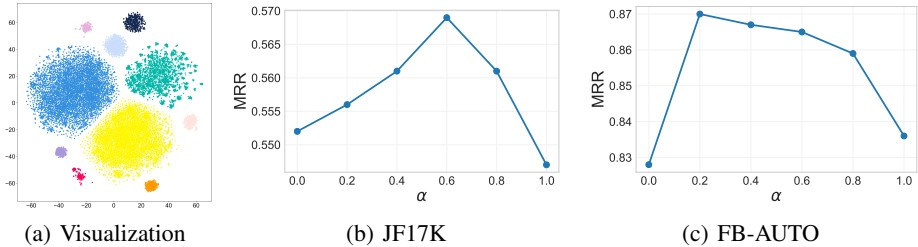

| (a) Visualization | (b) JF17K | (c) FB-AUTO |
|---|---|---|

Figure 4: (a) The visualization of mediator entity embeddings with top ten primary relations in WikiPeople via t-SNE; The effects of share embedding ratio $\alpha$ on (b) JF17K and (c) FB-AUTO.

We further investigate the effect of sharing hyperparameter $\alpha$ in Figure 4(b) and (c). An extreme point can be observed in both datasets, which estimates the semantic relatedness in corresponding datasets. Moreover, a higher sharing ratio $\alpha$ brings fewer model parameters, e.g., $\alpha = 0$ corresponds to the case that each mediator entity has independent embedding while $\alpha = 1$ means all mediator entities with the same primary relation own the same representation. Thus, a tradeoff between model parameter complexity and practical performance can be achieved.

## 5 CONCLUSION

In this paper, we propose TransEQ for HKG modeling. With the equivalent transformation developed, TransEQ successfully transforms a HKG to a KG without information loss. Especially, TransEQ builds the generalized encoder-decoder framework, which firstly captures both structural information and semantic information for HKG. Experiment results show that TransEQ obtains the state-of-the-art results on benchmark datasets. For future work, we would like to make the transformation design automated, such that each hyper-relational fact can be automatically transformed into a multi-relational subgraph following relation-specific transformation. Moreover, we plan to introduce specific GNN modules on the transformed KG to process the attribute information as well as primary information attached on the mediator entities.

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

CONTENTS

## A  OTHER RELATED WORK

### A.1  KNOWLEDGE GRAPH (KG) MODELING

Learning representations for entities and relations in KGs has been investigated thoroughly (Ji et al., 2021; Wang et al., 2017), which designs various SFs to model the semantics in triple knowledge $(s, r, o)$. Based on translational thought, TransE (Bordes et al., 2013), TransH (Wang et al., 2014) and RotatE (Sun et al., 2019) measure the distance between subject and object entities in a relation-specific latent space. Besides, ConvE (Dettmers et al., 2018) adopts the convolutional neural networks for SF design. TuckER (Balazevic et al., 2019) employs Tucker decomposition for SF design. Furthermore, several models combine the bilinear product with various types of embeddings (Cao et al., 2021; Trouillon et al., 2016; Yang et al., 2015). For example, ComplEx (Trouillon et al., 2016) and DulE (Cao et al., 2021) employ complex-valued embeddings and dual quaternion embeddings, respectively. However, models above ignore the multi-relational graph structure of KGs.

Until the emergence of message passing mechanism with GNN, structural information capture becomes an important topic in KG modeling. An encoder-decoder framework is developed in recent KG-based GNN studies, where GNNs encode structural information of KG and various SFs are combined for semantic information. Specifically, both R-GCN (Schlichtkrull et al., 2018) and SACN (Shang et al., 2019) treat the multi-relational KG as multiple single-relational graphs, and apply relational graph convolutional network (GCN) for entity representations. Moreover, VR-GCN (Ye et al., 2019) combines the translational idea with GNN to learn both entity and relation representations. CompGCN (Vashishth et al., 2019) develops three entity-relation composition operators to update entity representations in GCN, and KE-GCN (Yu et al., 2021) further incorporates the composition with relation update. NBFNet (Zhu et al., 2021) and RED-GNN (Zhang & Yao, 2022) also explore GNN with subgraph for KG completion. Overall, GNN-based models achieve promising results in KG modeling, which demonstrates the importance of capturing structural information.

### A.2  HYPERGRAPH & HYPEREDGE EXPANSION

A hypergraph is a generalization of graph, where a hyperedge can join any number of nodes (Ouvrard, 2020). Especially, hyperedge expansion (Agarwal et al., 2006; Dong et al., 2020; Zhou et al., 2006) is introduced to transform a hypergraph to a homogeneous graph, such that graph learning methods can work on hypergraphs (Feng et al., 2019; Yadati et al., 2019). Since HKG is viewed as a multi-relational ordered hypergraph (Yadati, 2020), here we investigate the representative expansion strategy of star expansion for additional insights.

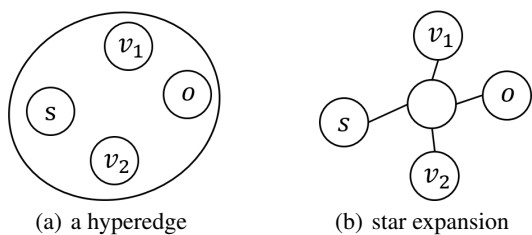

(a) a hyperedge          (b) star expansion

Figure 5: The illustration of star expansion on a hyperedge.

Figure 5 presents an example of a hyperedge with four nodes and its transformed graph by star expansion[2]. Specifically, for a hyperedge, the star expansion introduces a mediator node (like the blank node in the center of Figure 5(b)), which is then connected with all original nodes in the hyperedge. With the elegant transformation, hyperedge expansion has been widely applied in recommender systems (Xia et al., 2021), link prediction (Sun et al., 2021), etc.

On the other hand, the structural information loss has always been a concerned issue with hyperedge expansion strategy (Arya et al., 2021; Dong et al., 2020; Zhou et al., 2006). To be specific, an expansion strategy on hypergraph suffers from structural information loss, if there can be two distinct hypergraphs on the same node set reduced to the same graph by the expansion (Dong et al., 2020).

---

[2]The expansion strategy is named according to its graph illustration.

According to (Arya et al., 2021; Dong et al., 2020), the star expansion preserves the complete structural information. However, such traditional hyperedge expansion strategy cannot handle the HKG with hyper-relational semantics considered, which also guides our research that both structural and semantic information loss should be concerned in transforming a HKG to a KG.

## B    METHOD DETAILS

### B.1    OTHER VARIANTS OF TRANSFORMATIONS

To demonstrate the effectiveness of our proposed equivalent transformation in Section 3.1.1, here we further show other variants of transformations in Figure 6. Especially, the plain transformation in Figure 6(a) follows star expansion without attributes considered.

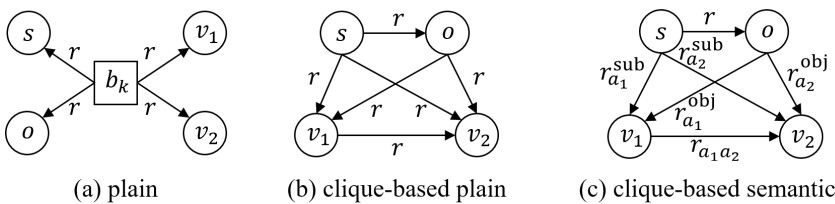

| (a) plain | (b) clique-based plain | (c) clique-based semantic |

Figure 6: The illustration of other variants of transformations.

In comparison to the star expansion, clique expansion is also a popular hyperedge expansion strategy (Dong et al., 2020; Zhou et al., 2006), which transforms the hyperedge into a clique subgraph, i.e., each pair of nodes in the hyperedge are connected in the transformed graph. Thus, we also extend clique expansion into the HKG case, e.g., the clique-based plain transformation in Figure 6(b) with only primary relations considered. To model the attribute information, in Figure 6(c), for each attribute $a_i$, the clique-based semantic transformation decomposes it into two relations of $r_{a_i}^{\text{sub}}$ and $r_{a_i}^{\text{obj}}$, which connect the value entity with subject and object entities, respectively. Each pair of value entities are also connected by devised relations between attributes to satisfy the clique structure. However, these variants of transformations bring information loss while our proposed equivalent one preserves complete information, as validated by both theoretical proof and experimental performance later.

### B.2    TRAINING PROCEDURE

---

**Algorithm 2:** TransEQ training algorithm.

---

**Input:** HKG $\mathcal{G}^H = (\mathcal{E}, \mathcal{R}, \mathcal{F}^H)$;
**Init:** $\boldsymbol{E}$ for $e \in \mathcal{E}$, $\boldsymbol{R}$ for $r \in \mathcal{R}$, $\boldsymbol{\theta}_{\text{Enc}}$ for GNN-based encoder, $\boldsymbol{\theta}_{\text{Dec}}$ for SF-based decoder;

1   Build encoder module $Enc()$ with $\boldsymbol{\theta}_{\text{Enc}}$;
2   Build decoder module $Dec()$ with $\boldsymbol{\theta}_{\text{Dec}}$;
3   Transform HKG $\mathcal{G}^H$ to KG $\mathcal{G}$ with Algorithm 1;
4   **for** $t = 1, \cdots, n_{\text{iter}}$ **do**
5      Sample a mini-batch $\mathcal{F}_{\text{batch}} \in \mathcal{F}^H$ of size $m_b$, $\mathcal{L} \leftarrow 0$;
6      $\boldsymbol{E}, \boldsymbol{R} = Enc(\mathcal{G}, \boldsymbol{E}, \boldsymbol{R}, \boldsymbol{\theta}_{\text{Enc}})$;
7      **for** $x := (s, r, o, \{(a_i, v_i)\}_{i=1}^n) \in \mathcal{F}_{\text{batch}}$ **do**
8          Construct negative samples $\mathcal{N}_x$;
9          $\phi(x) = Dec(x, \boldsymbol{E}, \boldsymbol{R}, \boldsymbol{\theta}_{\text{Dec}})$;
10         $\phi(x') = Dec(x', \boldsymbol{E}, \boldsymbol{R}, \boldsymbol{\theta}_{\text{Dec}}), \forall x' \in \mathcal{N}_x$;
11         Update loss $\mathcal{L} \leftarrow \mathcal{L} + \mathcal{L}_x(\phi)$ with $\mathcal{L}_x$ in equation 1;
12      **end**
13      Update learnable parameters w.r.t. the gradients $\nabla \mathcal{L}$;
14   **end**
**Output:** Embeddings $\boldsymbol{E}, \boldsymbol{R}$ and parameters $\boldsymbol{\theta}_{\text{Enc}}, \boldsymbol{\theta}_{\text{Dec}}$.

---

## C  THEORETICAL DETAILS

### C.1  SCORING FUNCTION COMPARISON

In Table 4, we present a comparison of representative HKG modeling studies with SF. Especially, TransEQ can leverage any SF therein for SF-based decoder.

Table 4:  A comparison of representative HKG modeling studies with scoring function. $\boldsymbol{p}_i$ is the position embedding in G-MPNN. Conv: convolutional neural network, FCN: fully connected network, Trf: Transformer, $[\cdot;\cdot]$: vector concatenation, Mean$(\cdot)$: element-wise average. $\min(\cdot)$: element-wise minimization.

| Model | Scoring Function |
|---|---|
| NaLP (Guan et al., 2019) | $\text{FCN}(\min_d(\text{Conv}([[\boldsymbol{h}_{r_s};\boldsymbol{h}_s];[\boldsymbol{h}_{r_o};\boldsymbol{h}_o];[\boldsymbol{h}_{a_i};\boldsymbol{h}_{v_i}]])))$ |
| m-DistMult (Fatemi et al., 2020) | $\langle\boldsymbol{h}_r,\boldsymbol{h}_s,\boldsymbol{h}_o,\boldsymbol{h}_{v_1},\cdots,\boldsymbol{h}_{v_n}\rangle$ |
| HypE (Fatemi et al., 2020) | $\langle\boldsymbol{h}_r,\text{Conv}(\boldsymbol{h}_s),\text{Conv}(\boldsymbol{h}_o),\text{Conv}(\boldsymbol{h}_{v_1}),\cdots,\text{Conv}(\boldsymbol{h}_{v_n})\rangle$ |
| HINGE (Rosso et al., 2020) | $\text{FCN}(\min_d([\text{Conv}([\boldsymbol{h}_r;\boldsymbol{h}_s;\boldsymbol{h}_o]);\text{Conv}([\boldsymbol{h}_r;\boldsymbol{h}_s;\boldsymbol{h}_o;\boldsymbol{h}_{a_i};\boldsymbol{h}_{v_i}])]))$ |
| G-MPNN (Yadati, 2020) | $\langle\boldsymbol{h}_r,\boldsymbol{p}_1,\cdots,\boldsymbol{p}_{n+2},\boldsymbol{h}_s,\boldsymbol{h}_o,\boldsymbol{h}_{v_1},\cdots,\boldsymbol{h}_{v_n}\rangle$ |
| StarE (Galkin et al., 2020) | $\boldsymbol{h}_o^\top\text{FCN}(\text{Mean}(\text{Trf}(\boldsymbol{h}_r,\boldsymbol{h}_{a_1},\cdots,\boldsymbol{h}_{a_n},\boldsymbol{h}_s,\boldsymbol{h}_{v_1},\cdots,\boldsymbol{h}_{v_n})))$ |
| TransEQ | Arbitrary SF |

### C.2  COMPLEXITY ANALYSIS ON TRANSFORMATIONS

Based on the description in Section B.1, we analyze the parameter complexity of different transformations in Table 5.

Table 5: The parameter complexity of different transformations, in terms of entity/node, relation and edge. $n_e = |\mathcal{E}|$ and $n_r = |\mathcal{R}|$ are the number of entities and relations in HKG. $n_r^{\text{pri}}$ and $n_r^{\text{qua}}$ are the numbers of primary and qualifier relations, respectively. $n_a$ is the maximum number of attribute-value qualifiers for facts. $N^{\text{qua}}$ and $N^{\text{pri}}$ are the number of hyper-relational facts with and without attribut-value qualifiers, such that $N^{\text{pri}} + N^{\text{qua}} = |\mathcal{F}|$.

| Transformation | $\mathcal{O}_{\text{ent}}/\mathcal{O}_{\text{node}}$ | $\mathcal{O}_{\text{edge}}$ | $\mathcal{O}_{\text{rel}}$ |
|---|---|---|---|
| plain | $\mathcal{O}(n_e+N^{\text{qua}})$ | $\mathcal{O}(N^{\text{pri}}+N^{\text{qua}}(n_a+2))$ | $\mathcal{O}(n_r^{\text{pri}})$ |
| clique-based plain | $\mathcal{O}(n_e)$ | $\mathcal{O}(N^{\text{pri}}+N^{\text{qua}}(n_a+2)^2)$ | $\mathcal{O}(n_r^{\text{pri}})$ |
| clique-based semantic | $\mathcal{O}(n_e)$ | $\mathcal{O}(N^{\text{pri}}+N^{\text{qua}}(n_a+2)^2)$ | $\mathcal{O}(n_r^{\text{pri}}+n_a n_r^{\text{qua}})$ |
| equivalent | $\mathcal{O}(n_e+N^{\text{qua}})$ | $\mathcal{O}(N^{\text{pri}}+N^{\text{qua}}(n_a+3))$ | $\mathcal{O}(3n_r^{\text{pri}}+n_r^{\text{qua}})$ |

According to the transformation design, clique-based transformations introduce pairwise edges for relatedness while star-based ones of plain transformation and equivalent transformation rely on additional mediator entities. Therefore, clique-based transformations keep the node complexity of $\mathcal{O}(n_e)$ while star-based ones build $\mathcal{O}(n_e+N^{\text{qua}})$ nodes. On the other hand, the plain transformation keeps the same edge complexity with the original HKG structure, while a relational edge between subject and object entities is added in equivalent transformation for semantic difference, bringing the complexity increase of $\mathcal{O}(N^{\text{qua}})$. Compared with the relation complexity of about $\mathcal{O}(n_r^{\text{pri}}+n_r^{\text{qua}})$ in the original HKG, the equivalent transformation introduces $\mathcal{O}(n_r^{\text{pri}})$ relations to distinguish links between subject and object entities, which are acceptable in practice.

### C.3  PROOF OF INFORMATION PRESERVATION

To demonstrate the zero information loss in the equivalent transformation, in Algorithm 3, we present the process that can equivalently recover the original HKG from the transformed KG.

Note that $\mathcal{N}_{b_k}$ in line 3 is a subgraph, and attribute-value qualifiers can be extracted from direct relational links to mediator $b_k$ in line 5. Here we consider hyper-relational fact with at least one

qualifier, while triple facts can be directly added in recovered HKG due to no mediator. Thus, Algorithm 1 and Algorithm 3 form an equivalent conversion between HKG and KG, i.e., the equivalent transformation preserves the complete information. In comparison, the plain transformation and clique-based plain transformation only keep primary relations in conversion, which cannot be recovered due to attribute loss. Besides, clique-based semantic transformation inherits the structural information loss of clique expansion in hyperedge expansion (Dong et al., 2020).

---

**Algorithm 3:** The algorithm for recovering HKG from the transformed KG by equivalent transformation.

---

**Input:** Transformed KG $\mathcal{G} = (\mathcal{E}, \mathcal{R}, \mathcal{F})$;
**Init:** Recovered HKG $\mathcal{G}^H = (\mathcal{E}^H, \mathcal{R}^H, \mathcal{F}^H)$ with $\mathcal{E}^H \leftarrow \emptyset, \mathcal{R}^H \leftarrow \emptyset, \mathcal{F}^H \leftarrow \emptyset$;

1   Obtain the set of mediator entities from $\mathcal{E}, \mathcal{E}^{\mathrm{med}}$;
2   **for** $b_k \in \mathcal{E}^{\mathrm{med}}$ **do**
3      Find $b_k$'s neighbor entities and their connected relations from $\mathcal{F}$, $\mathcal{N}_{b_k} = \{r_i, e_i\}_{i=1}^n$;
4      Extract $(s, r, o)$ from $\mathcal{N}_{b_k}$ via motif-structure discovery;
5      Extract $\{(a_i, v_i)\}_{i=1}^{n-2}$ from left parts of $\mathcal{N}_{b_k}$;
        `// part of` $\{r_i, e_i\}_{i=1}^n$ `corresponds to` $\{(a_i, v_i)\}_{i=1}^{n-2}$.
6      $\mathcal{E}^H \leftarrow \mathcal{E}^H \cup \{e_i\}_{i=1}^n, \mathcal{R} \leftarrow \mathcal{R} \cup \{r\} \cup \{a_i\}_{i=1}^{n-2}$;
7      $\mathcal{F}^H \leftarrow \mathcal{F}^H \cup \{(s, r, o, \{(a_i, v_i)\}_{i=1}^{n-2})\}$;
8   **end**
**Output:** Recovered HKG $\mathcal{G}^H = (\mathcal{E}^H, \mathcal{R}^H, \mathcal{F}^H)$.

---

### C.4   PROOF OF FULL EXPRESSIVITY

Our proposed TransEQ firstly transforms a HKG to a KG, then develops a GNN-based encoder for representation encoding, and calculates plausibility scores based on existing SFs in HKG modeling studies with entity and relation embeddings from encoder part. Meanwhile, several SFs from HypE (Fatemi et al., 2020), BoxE (Abboud et al., 2020), RAM (Liu et al., 2021), etc., have been proved to be fully expressive with an assignment of entity and relation embeddings in their original papers. Hence, with a fully expressive SF in decoder, the TransEQ model is fully expressive if the output embeddings from encoder part follow corresponding assignment required by SF, which is proved as follows, *Proof.* For $t \in \mathcal{E}, r \in \mathcal{R}$, let $\boldsymbol{h}_t^0, \boldsymbol{h}_r^0$ denote their initialized representations, while $\boldsymbol{h}_t^L, \boldsymbol{h}_r^L$ denote corresponding embeddings outputted from encoder part. We also denote $\boldsymbol{h}_t^{\mathrm{SF}}, \boldsymbol{h}_r^{\mathrm{SF}}$ the required input embeddings of SF in decoder. Then, in mathematical, with $\boldsymbol{h}_t^L = Enc(\boldsymbol{h}_t^0, \boldsymbol{\theta}_{\mathrm{Enc}})$ and $\boldsymbol{h}_r^L = Enc(\boldsymbol{h}_r^0, \boldsymbol{\theta}_{\mathrm{Enc}})$, we should prove $\boldsymbol{h}_t^L = \boldsymbol{h}_t^{\mathrm{SF}}$ and $\boldsymbol{h}_r^L = \boldsymbol{h}_r^{\mathrm{SF}}$ can be achieved with appropriate choice of encoder parameters $\boldsymbol{\theta}_{\mathrm{Enc}}$[3] and initialized embeddings $\boldsymbol{h}_t^0, \boldsymbol{h}_r^0$.

Taking the example of R-GCN (Schlichtkrull et al., 2018) as encoder, the message passing process of each GCN layer can be written as,

$$\boldsymbol{h}_t^{l+1} = \sigma\Big(\sum_{r \in \mathcal{R}} \sum_{u \in \mathcal{N}_t^r} \frac{1}{|\mathcal{N}_t^r|} \boldsymbol{W}_r^l \boldsymbol{h}_u^l + \boldsymbol{W}_0^l \boldsymbol{h}_t^l\Big),$$

where $\sigma$ denotes nonlinear activation function like ReLU, which is unnecessary and can be removed (Wu et al., 2019).

Now, we describe a feasible assignment of encoder parameters: For each layer $l \in \{1, \cdots, L\}$ and $r \in \mathcal{R}$, relation-specific matrix $\boldsymbol{W}_r^l$ is set to null matrix, while $\boldsymbol{W}_0^l$ is set to identity matrix, where both $\boldsymbol{W}_r^l$ and $\boldsymbol{W}_0^l$ belong to encoder parameters $\boldsymbol{\theta}_{\mathrm{Enc}}$.

Following the assignment above, we have $\boldsymbol{h}_t^{l+1} = \boldsymbol{h}_t^l$, i.e., $\boldsymbol{h}_t^L = \boldsymbol{h}_t^0$. Hence, we can set the values of $\boldsymbol{h}_t^0$ according to $\boldsymbol{h}_t^{SF}$. In R-GCN, $\boldsymbol{h}_r^L$ is directly initialized and can be set to $\boldsymbol{h}_r^{\mathrm{SF}}$. Overall, the encoder's output embeddings follow the required embedding assignment of SF with above assignment on $\boldsymbol{\theta}_{\mathrm{Enc}}, \boldsymbol{h}_t^0$ and $\boldsymbol{h}_r^0$. Thus, the expressivity of TransEQ is proved to be in accord with that of the SF it uses in decoder. Finally, we note that the proof can be trivially extended to other GNN-based encoders like CompGCN (Vashishth et al., 2019) by introducing extra assignments on encoder parameters. □

---

[3]Note that here we simplify the expression of encoder module, which is still in accord with the form in Algorithm 2.

## D    EXPERIMENT DETAILS

Here we provide more experiment details to support our claim. Moreover, we further perform experiments on five datasets to validate the robustness of our proposed TransEQ model.

### D.1    DATASET DETAILS

We detail the dataset statistics in Table 6.

Table 6: Dataset statistics.

| Dataset | $|\mathcal{E}|$ | $|\mathcal{R}|$ | #Train | #Valid | #Test |
|---|---|---|---|---|---|
| WikiPeople | 47,765 | 707 | 305,725 | 38,223 | 38,281 |
| JF17K | 28,645 | 322 | 61,104 | 15,275 | 24,568 |
| FB-AUTO | 3,388 | 8 | 6,778 | 2,255 | 2,180 |

To validate the robustness, we further consider a recently developed dataset WD50K and its variant WD50K(100) (Galkin et al., 2020), where all facts contain qualifiers, i.e., no simple triple facts therein. We also consider four datasets of WikiPeople-3, JF17K-3, WikiPeople-4, JF17K-4, developed from (Liu et al., 2020), where facts have a fixed number of qualifiers in accord with the dataset name. Table 7 presents dataset statistics.

Table 7: Dataset statistics.

| Dataset | $|\mathcal{E}|$ | $|\mathcal{R}|$ | #Train | #Valid | #Test |
|---|---|---|---|---|---|
| WD50K | 47,156 | 532 | 166,435 | 23,913 | 46,159 |
| WD50K(100) | 18,792 | 279 | 22,738 | 3,279 | 5,297 |
| WikiPeople-3 | 12,270 | 66 | 20,656 | 2,582 | 2,582 |
| JF17K-3 | 11,541 | 104 | 27,635 | 3,454 | 3,455 |
| WikiPeople-4 | 9,528 | 50 | 12,150 | 1,519 | 1,519 |
| JF17K-4 | 6,536 | 23 | 7,607 | 951 | 951 |

### D.2    IMPLEMENTATION DETAILS

We implement TransEQ in PyTorch (Paszke et al., 2019) with Adam optimizer. The embedding dimension $d$ is set to the typical size 200 (Abboud et al., 2020; Fatemi et al., 2020; Galkin et al., 2020; Wang et al., 2021; Yu & Yang, 2021). The batch size, learning rate and dropout are chosen from {64, 128}, {0.0001, 0.0005, 0.001, 0.005} and [0.1, 0.5] with step 0.1, respectively. Besides, we mainly adopt CompGCN (Vashishth et al., 2019) as encoder and m-DistMult (Fatemi et al., 2020) as decoder. For the encoder part, the number of GNN layers and sharing ratio $\alpha$ are chosen from {1, 2, 3, 4} and [0.0, 1.0] with step 0.2, respectively. The composition operation in encoder is set to rotate function (Sun et al., 2019). We tune hyperparameters over the validation set with early stopping strategy employed. All experiments are run on a RTX 2080 Ti GPU.

### D.3    EFFICIENCY COMPARISON

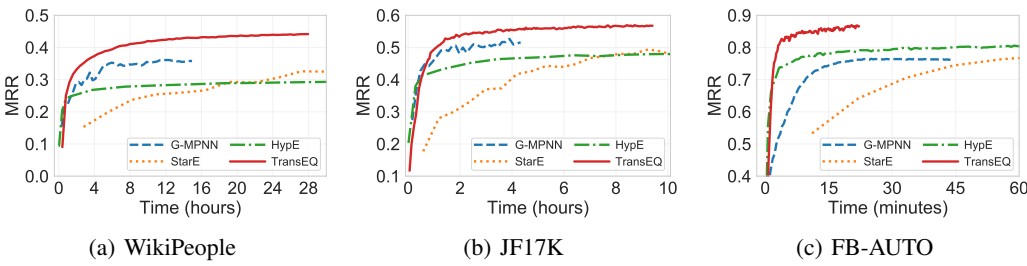

(a) WikiPeople          (b) JF17K          (c) FB-AUTO

Figure 7: Comparison on clock time of model training vs. testing MRR.

Furthermore, we compare the learning processes of TransEQ with structural modeling approaches on three datasets in Figure 7. The learning curve of HypE with linear time complexity is also plotted for comparison. It can be observed that TransEQ achieves similar convergence speed with HypE in practice, which owes to the multilinear product based SF (Liu et al., 2021) and efficient implementation. With a similar form of SF adopted, G-MPNN achieves a close convergence rate but inferior performance, which demonstrates the strength of GNN-based encoder compared with HGNN. As for StarE with Transformer-based SF, tremendous parameters lead to time-consuming training on all datasets.

## D.4 TRANSFORMATION COMPARISON

To analyze the effects of various transformations, we present the performance comparison in Table 8. Due to space limitation, results with Hit@3 are omitted, which are in accord with other metrics. As described in Section 3.1.1, our proposed equivalent transformation connects subject and object entities via a relational edge $r$ to form the motif for semantic difference. Thus, we investigate the effectiveness of such operation by removing the edge in the transformation, referred to as w/o distinction transformation.

Table 8: Performance comparison of different transformations. Best results are highlighted in bold.

| Transformation | JF17K | | | FB-AUTO | | |
|---|---|---|---|---|---|---|
| | MRR | Hit@1 | Hit@10 | MRR | Hit@1 | Hit@10 |
| plain | 0.504 | 0.422 | 0.669 | 0.836 | 0.809 | 0.884 |
| clique-based plain | 0.557 | 0.478 | 0.706 | 0.824 | 0.804 | 0.859 |
| clique-based semantic | 0.552 | 0.474 | 0.706 | 0.831 | 0.809 | 0.869 |
| w/o distinction | 0.536 | 0.456 | 0.695 | 0.849 | 0.810 | 0.884 |
| equivalent | **0.569** | **0.489** | **0.722** | **0.870** | **0.842** | **0.909** |

From the table, we can observe that the equivalent transformation outperforms other variants of transformations, which is in accord with the information loss analysis in Section 3.2, i.e., only equivalent transformation preserves complete information. Moreover, removing the relational edge in the equivalent transformation leads to a Hit@1 performance drop of 7% on JF17K, which demonstrates the effectiveness and necessity of considering semantic difference in the transformation. Besides, since any two entities are connected in two clique-based transformations, the relatedness between entities is largely captured and thus they obtain close performance, i.e., the clique structure makes these transformations insensitive to semantic information. In comparison, the star structure in plain transformation and equivalent transformation is quite simple, and additional information including hyper-relational semantics and semantic difference should be incorporated in transformation, which also accounts for the obvious gap between these two transformations. Considering the zero information loss and experimental performance, the equivalent transformation becomes the best choice for TransEQ in HKG modeling.

## D.5 ENCODER-DECODER CHOICE COMPARISON

We also compare the performance of different encoder-decoder choices of TransEQ models on FB-AUTO in Table 9. These results further validate the observations in Section 4.3.

## D.6 ADDITIONAL HKG COMPLETION RESULTS

Since on WD50K and WD50K(100) former studies (Galkin et al., 2020; Yu & Yang, 2021) only predict missing entities at primary triple, not comparable to HKG completion task in Section 4.1, we select competitive baselines in Table 2 and report their performance on these datasets, as shown in Table 10. Here we evaluate TransEQ models with m-DistMult and Transformer as decoders, denoted by TransEQ-DM and TransEQ-Trf, respectively. According to the table, TransEQ model with Transformer-based decoder generally performs well on both datasets, which again demonstrates the effectiveness of model design.

In Table 11, with m-DistMult and HypE(HP) as decoders, we further investigate TransEQ's performance on HKG datasets with fixed number of qualifiers, compared with HINGE (Rosso et al., 2020),

Table 9: Performance comparison of different encoder-decoder choices on FB-AUTO. "OOM" indicates out of memory. Best results for each encoder are highlighted in bold.

| Encoder X → | FB-AUTO | | | | | |
| | CompGCN | | R-GCN | | No Encoder | |
| Decoder Y ↓ | MRR | Hit@10 | MRR | Hit@10 | MRR | Hit@10 |
|---|---|---|---|---|---|---|
| m-TransH | 0.825 | 0.873 | OOM | | 0.728 | 0.728 |
| Transformer | 0.846 | 0.899 | OOM | | **0.834** | **0.897** |
| m-DistMult | **0.870** | **0.909** | 0.834 | 0.892 | 0.784 | 0.845 |
| HypE | 0.860 | 0.902 | **0.840** | **0.892** | 0.804 | 0.856 |

Table 10: Results of HKG completion on WD50K(100) and WD50K. Best results are highlighted in bold. "-" denotes exceeding time limit.

| Model | WD50K(100) | | | WD50K | | |
| | MRR | Hit@1 | Hit@10 | MRR | Hit@1 | Hit@10 |
|---|---|---|---|---|---|---|
| NeuInfer | 0.289 | 0.252 | 0.358 | 0.179 | 0.141 | 0.250 |
| HypE | 0.436 | 0.339 | 0.617 | 0.233 | 0.158 | 0.377 |
| BoxE | **0.672** | **0.609** | 0.747 | - | - | - |
| TransEQ-DM | 0.553 | 0.516 | 0.621 | 0.290 | 0.232 | 0.400 |
| TransEQ-Trf | 0.661 | 0.608 | **0.756** | **0.343** | **0.278** | **0.465** |

NeuInfer (Guan et al., 2020), n-TuckER (Liu et al., 2020) and GETD (Liu et al., 2020). Note that n-TuckER and GETD can only handle datasets with a fixed number of qualifiers with competitive performance. According to the results, TransEQ models obtain the state-of-the-art performance on most datasets, indicating the robustness and effectiveness of the proposed equivalent transformation as well as the generalized encoder-decoder framework.

Table 11: Results of HKG completion on datasets with fixed number of attribute-value qualifiers. Results of baselines are collected from original papers and (Di & Chen, 2022; Liu et al., 2020).

| Model | WikiPeople-3 | | JF17K-3 | | WikiPeople-4 | | JF17K-4 | |
| | MRR | H@10 | MRR | H@10 | MRR | H@10 | MRR | H@10 |
|---|---|---|---|---|---|---|---|---|
| HINGE | 0.338 | 0.508 | 0.587 | 0.738 | 0.352 | 0.557 | 0.745 | 0.842 |
| NeuInfer | 0.355 | 0.521 | 0.622 | 0.770 | 0.361 | 0.566 | 0.765 | 0.871 |
| n-TuckER | 0.373 | 0.558 | 0.727 | 0.852 | 0.362 | 0.570 | 0.804 | 0.902 |
| GETD | 0.373 | **0.558** | **0.732** | **0.856** | 0.386 | 0.596 | 0.810 | 0.910 |
| TransEQ-DM | **0.382** | 0.557 | 0.685 | 0.827 | 0.378 | 0.614 | **0.820** | **0.923** |
| TransEQ-HP | 0.370 | 0.557 | 0.726 | 0.847 | **0.394** | **0.602** | 0.805 | 0.908 |

## D.7 ADDITIONAL VISUALIZATION RESULTS

Following the settings in Section 4.4, in Figure 8, we compare the visualization results of utilizing independent embedding ($\alpha = 0.0$) and sharing embedding (the best setting with $\alpha = 0.8$), which further validates the effectiveness of TransEQ capturing semantic relatedness for mediator entities.

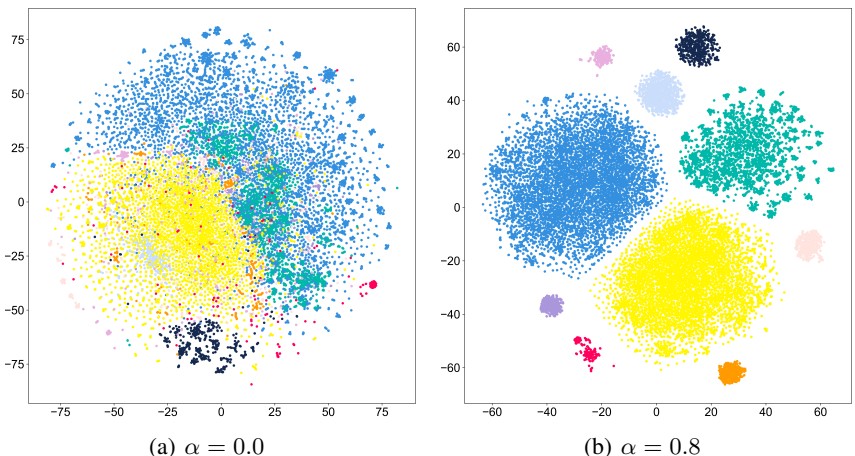

(a) $\alpha = 0.0$        (b) $\alpha = 0.8$

Figure 8: The visualization of mediator entity embeddings with top ten primary relations in WikiPeople via t-SNE on share embedding ratio (a) $\alpha = 0.0$ and (b) $\alpha = 0.8$.

