# OpenReview forum: "Two Birds, One Stone: An Equivalent Transformation for Hyper-relational Knowledge Graph Modeling"
_ICLR.cc/2023/Conference — Submitted to ICLR 2023_

### Official Review · Reviewer_FnvL · 2022-10-21

**Confidence:** 5
**Correctness:** 4
**Technical Novelty And Significance:** 3
**Empirical Novelty And Significance:** 3
**Recommendation:** 8

**Clarity, Quality, Novelty And Reproducibility:**

The paper is well-written and easy to follow. The code is also available anonymously for reproducibility.

**Strength And Weaknesses:**

This paper studies the important problem of modeling hyper-relational knowledge graphs.

Despite the simplicity, the proposed approach in the paper achieves state-of-the-art experimental results and strong theoretical properties.

The experimental results are reported on various datasets with different statistics.

**Summary Of The Paper:**

This paper proposes TransEQ: a method for hyper-relational knowledge graph embedding. The motivation behind TransEQ is considering both semantic and structural information in a hyper-relational knowledge graph. The proposed method achieves state-of-the-art results on established datasets and strong theoretical properties.


**Summary Of The Review:**

The method used in the paper to transform an HKG into a KG is almost identical to the reification method discussed in [1]. Fatemi et al [1] showed that transforming the HKG into a KG (conversion approaches) using reification (or S2C) and applying SFs does not work. It is exciting to see that transforming an HKG into a KG and using GNNs+FSs yield better results. I appreciate it if the authors can add detail about this to their paper.

I do not agree with the paper on the importance of semantic information in an HKG. The paper implies that for a hyper-relation educated at, the school is more important than the degree that the corresponding person is educated. This is, however, not given in many datasets and might vary from one dataset to another. Regarding this, can the authors elaborate on how the primary triple in a hyper-relation is identified? Especially for the datasets used in this paper.

[1] Fatemi, Bahare, Perouz Taslakian, David Vazquez, and David Poole. "Knowledge hypergraphs: Prediction beyond binary relations."

---

> ### Author Response · Authors · 2022-11-19
> **Detailed Response to Reviewer FnvL**
>
> Thank you for your interest, positive comments on the significance, and kind words. Here are our responses to your concerns:
>
> **(1) Why GNN+SFs can yield better results.**
> As raised by the reviewer, simply applying S2C and SFs for HKG modeling does not work in (Fatemi et al., 2020), while we attribute the significant improvement of our proposed TransEQ model to both structural and semantic information captured in the encoder-decoder framework.
>
> The original implementation in (Fatemi et al., 2020) ignores the structural connections in converted graph, and only uses KG-based SF to model the semantic information. In comparison, we first use the equivalent transformation and GNN-based encoder to capture structural information in refied KG, and further adopt the advanced HKG-based SF to model the semantic information in decoder. The performance comparison of different encoder-decoder choices in Table 3 and Table 9 also validate the effectiveness of encoder design combined with traditional HKG models. We will add the details of making S2C work in the revision.
>
> **(2) About the importance of semantic information.**
>
> Thanks for the comments.
>
> Firstly, the importance of semantic information, i.e., the semantic difference (or the primary triplet v.s. qualifiers) is a commonly used concept in related studies of StarE (Galkin et al., 2020), HINGE (Rosso et al., 2020), NeuInfer (Guan et al., 2021), etc., which assume a hyper-relational fact is composed by a base triplet $(s,r,o)$ with a set of qualifiers $\lbrace(a_i,v_i)\rbrace^n_{i=1}$. The described example also follows the aforementioned studies.
>
> Secondly, we agree with the reviewer that the importance of semantic information varies from one dataset t another. However, for a hyper-relational fact, the semantic difference modeling is necessary, as validated in our ablation study above (The "Drop Original Triple" corresponds to the case without semantic difference considered).
> |                |       | JF17K |       |       | FB-AUTO |       |
> |:--------------:|:-----:|:-----:|:-----:|:-----:|:-------:|:-----:|
> |                |  MRR  | HR@10 |  HR@1 |  MRR  |  HR@10  |  HR@1 |
> | Drop Original  | 0.536 | 0.695 | 0.456 | 0.849 |  0.884  | 0.810 |
> |     TransEQ    | 0.569 | 0.722 | 0.489 | 0.870 |  0.909  | 0.842 |
>
> We will clarify the semantic difference in the revision, and identify its necessity in HKG modeling.
>
> Next, we introduce the identification of primary triple (following the setting in (Galkin et al., 2020; Rosso et al., 2020; Guan et al., 2021) ), mainly based on the samples in evaluated datasets.
>
> * **WikiPeople**
>     * Raw data format: $\lbrace r^{\text{sub}}: e_s, r^{\text{obj}}: e_o, r_1: e_1, \cdots, r_n: e_n\rbrace$
>     * Identification: Since the raw data provide the subject and object entities, we combine $r^{\text{sub}}$ and $r^{\text{obj}}$ to a new primary relation $r$.
>     * Converted data format: $(e_s, r, e_o, {\lbrace( r_i:e_i)\rbrace}^n_{i=1})$
> * **WikiPeople Example**
>     * Raw data sample: **{P3919\_h:Q337913, P3919\_t:Q1210343, P2868: Q864380}**
>     * Identification: The primary triple **(Q337913, P3919, Q1210343)**
>     * Converted data sample: **(Q337913, P3919, Q1210343, (P2868,Q864380))**
>     *
> * **JF17K/FB-AUTO**
>     * Raw data format: $(r, e_s, e_o, e_1, \cdots, e_n)$
>     * Identification: The first two entities in the raw data are default subject and object entities. We split the (n+2)-ary relation $r$ into $r^{so}, r_1, \cdots, r_n$.
>     * Converted data format: $(e_s, r^{so}, e_o, {\lbrace( r_i:e_i)\rbrace}^n_{i=1})$
> * **JF17K Example**
>     * Raw data sample: **(soccer.football_player_match_participation,	02pp1,	04xkpd,	0c0lv1g)**
>     * Identification: The primary triple **(02pp1, soccer.football_player_match_participation_so, 04xkpd)**
>     * Converted data sample: **(02pp1, soccer.football_player_match_participation_so, 04xkpd, (soccer.football_player_match_participation_1, 0c0lv1g))**

---

### Official Review · Reviewer_94ho · 2022-10-24

**Confidence:** 4
**Correctness:** 2
**Technical Novelty And Significance:** 2
**Empirical Novelty And Significance:** 2
**Recommendation:** 3

**Clarity, Quality, Novelty And Reproducibility:**

Clarity. Writing is understandable, but wordy and there are smaller language issues. Theorem 1 is trivial and not needed. Theorem 2 lacks a rigid definitions and is meaningless to me as written.

Novelty. Little (see W1).

Reproducability. Unclear (see W2).


**Strength And Weaknesses:**

Strengths:

S1. Simple approach (but see W1).

S2. Good experimental results (but see W2).

Weaknesses:

W1. Little novelty. The proposed transformation is equivalent to reification (e.g, as used in RDF, not discussed in paper) + keeping the original fact. The introduced blank nodes share (partial) embeddings with their subject entity, which is a good idea but also the immediate approach after a minute of thought. The GNNs being used are out-of-the-box relational GNNs.

W2. Experimental study not insightful. (i) The experimental study does not describe the experimental setup, most notably hyperparameter optimization. (ii) The proposed methods are sometimes close, sometimes further away in quality compared to related work, but it's not explored why. (iii) In the datasets used in the appendix, the proposed method seems to provide little benefits compared to prior work. (iv) The protocol for evaluating "link prediction" in the experimental study is not convincing: It uses a ranking protocol where only one of the entities in a hyperfact is hidden (e.g., degree) but the others are specified (e.g., university). This problem statement does not appear to be model a practical use case. How would one use such a method to predict a new fact and how good would it be? (v) There are no ablation studies to justify the modelling choices (e.g., just make everything binary, or do not keep the original triple after reification).

W3. Arguments about "semantic difference" unclear. For a fact such as "A graduated at B obtaining degree C", the paper argues that the "graduated at" relation is a primary relation, whereas the degree is auxiliary. This argument is not convincing (one may argue the other way around) to me, and, in fact, the is little justification of the need to capture "semantic difference". Moreover, even if reification is used (i.e., without keeping the original fact) the HKG is still reconstructable.


**Summary Of The Paper:**

Studies link prediction for hyper-relational knowledge graphs (in which edges are associated with have attribute/entity pairs). Transforms the hyper-relational KG into a standard KG using reification, then runs a relational GNN for link prediction.


**Summary Of The Review:**

Simple, potentially useful approach, but little novelty and not insightful.

---

> ### Author Response · Authors · 2022-11-19
> **Detailed Response to Reviewer 94ho (Part 1)**
>
> We want to thank the reviewer for their positive feedback and the extremely interesting questions raised. Specifically, we have answered the raised questions as follows:
>
> **(1) About the novelty.**
>
> The main novelty of this work is proposing a simple yet effective manually-designed transformation, which obtains surprisingly good results on HKG completion task. Owing to the designed transformation, the proposed TransEQ elegantly captures both structural information (hyper-relational graph structure) and semantic information (semantic difference) in HKG, while most existing works fail to do so, as shown in Table 1. Besides, the designed transformation transforms HKG into KG, which allows for a proper use of mature techniques developed for KG, i.e., encoder-decoder framework.
>
> Results in Table 2 and Table 3 also demonstrate the effectiveness of this simple yet effective design, as well as its generality in collaborating with a wide variety of encoder/decoder models.
>
> We believe the simple design with significant performance achieved is novel, contributing to the simplicity, which can bring valuable insights and rethinking discussions to the community.
>
> Whilst the KG-based models used in encoder and the HKG-based SFs used in decoder are from existing works, no works before ever tried to bridge the research in KG and HKG together. We believe the idea of using equivalent transformation to bridge both is novel.
>
> Moreover, the RDF raised by the reviewer has already been discussed in our paper above Section 3.1. We identify both RDF reification and CVT in Freebase as a motivation for our work transforming a HKG to a KG with the encoder-decoder framework combined.
>
> Besides, the idea might be got after a minute of thought (although we get this for a long time of investigation and thinking), but we think another important thing is to validate the idea, otherwise, the research area won't be moved forward.

---

> > ### Author Response · Authors · 2022-11-21
> > **Detailed Response to Reviewer 94ho (Part 2)**
> >
> >
> > **(2-i) The experiment setup.**
> >
> > We follow the typical experiment setup in HKG modeling research as referred in **Task and Evaluation Metrics** with (Abboud et al., 2020; Fatemi et al., 2020; Guan et al., 2019; Liu et al., 2020; Wang et al., 2021), and the implementation details are provided in Section D.2.
> >
> > **(2-ii) The close or far performance gap with baselines.**
> >
> > We copy experiment results of all baselines from their original paper or related papers, since related studies adopt the same benchmarks and dataset split, as described in Table 2.
> >
> > The performance gaps between our model and baselines on different datasets are mainly determined by the dataset characteristics and the absolute performance on that benchmark. For example, the SOTA MRR performance on WikiPeople is 0.395, while on FB-AUTO is 0.844. So the improvement space is larger for WikiPeople than FB-AUTO, which explains the large (0.059) or small (0.026) gap between our proposed model and baselines.
> >
> > Anyway, the experiment results indicate the necessity and value of a robust model like TransEQ for this research question.
> >
> > **(2-iii) The performance comparison in appendix.**
> >
> > We provide the additional experiment results in appendix to measure the robustness of our proposed TransEQ. As we can see in Table 10-11, TransEQ achieves better or comparable performance compared with baselines. Besides, the strong baseline of GETD can only apply to single-arity KG, which does not apply to HKG modeling. Moreover, the three datasets of WikiPeople, JF17K and FB-AUTO are commonly used benchmarks in this area (Abboud et al., 2020; Fatemi et al., 2020; Guan et al., 2019; Liu et al., 2020; Wang et al., 2021), on which TransEQ beats all baselines.
> >
> > **(2-iv) The evaulation protocol.**
> >
> > Again, we follow the typical experiment setup in HKG modeling research as referred in **Task and Evaluation Metrics** with (Abboud et al., 2020; Fatemi et al., 2020;
> > Guan et al., 2019; Liu et al., 2020; Wang et al., 2021).
> >
> > For the testing sample $x:=(s,r,o,{\lbrace(a_i,v_i)\rbrace}^n_{i=1})$, we assume that either entity in $s, o, v_1,\cdots,v_n$ may be missing, and apply the model to predict missing entity at each position in evaluation.
> >
> > Moreover, for the case of missing multiple entities, we can use a brute-force enumeration algorithm to infer the missing entity one by one, e.g., for $(s, r, ?, (a_1, ?))$ with two missing entities:
> > * For $e\in\mathcal{E}$ with $(s, r, e, (a_1, ?))$:
> >     * For $e^\prime\in\mathcal{E}$:
> >         * calculate score for $(s, r, e, (a_1, e^\prime))$
> > * Select the fact with maximum score as the missing fact.
> >
> > However, the proposed setting is a totally new setting that has not been explored in existing studies. Thanks for the suggestion, and we will try to evaluate it in future work.
> >
> > **(2-v) There is no ablation study.**
> >
> > The reviewer can refer to B.1 and Table 8 for the ablation study of the equivalent transformation, e.g., Figure 6.(a) corresponds to only keep primary relaiton, and Figure 6.(b)-(c) correspond to othe types of reification.
> > Besides, the ablation study of not keeping the original triple after reification is as follows:
> > |                |       | JF17K |       |       | FB-AUTO |       |
> > |:--------------:|:-----:|:-----:|:-----:|:-----:|:-------:|:-----:|
> > |                |  MRR  | HR@10 |  HR@1 |  MRR  |  HR@10  |  HR@1 |
> > | Drop Original  | 0.536 | 0.695 | 0.456 | 0.849 |  0.884  | 0.810 |
> > |     TransEQ    | 0.569 | 0.722 | 0.489 | 0.870 |  0.909  | 0.842 |
> >
> > Moreover, the results in Table 3 and Table 9 provide the ablation study of encoder-decoder framework in TransEQ.

---

> > > ### Author Response · Authors · 2022-11-21
> > > **Detailed Response to Reviewer 94ho (Part 3)**
> > >
> > >
> > > **(3-i) About "semantic difference".**
> > >
> > > Firstly, the semantic difference (or the primary triplet v.s. qualifiers) is a commonly used concept in related studies of StarE (Galkin et al. 2020), HINGE (Rosso et al. 2020), NeuInfer (Guan et al. 2021), etc., which assume a hyper-relational fact is composed by a base triplet $(s,r,o)$ with a set of qualifiers $\lbrace(a_i,v_i)\rbrace^n_{i=1}$.
> > >
> > > Secondly, we agree with the reviewer that semantic difference depends on the need to capture, and may change in different cases. However, for a hyper-relational fact, the semantic difference modeling is necessary, as validated in our ablation study above (The "Drop Original Triple" corresponds to the case without semantic difference considered).
> > >
> > > We will clarify the semantic difference in the revision, and identify its necessity in HKG modeling.
> > >
> > > **(3-ii) The reconstruction of reification.**
> > >
> > > The reification is not a reconstructable transformation for HKG.
> > >
> > > For example, given the $k$-th hyper-relational fact (*Alan Turing*, educated at, *Cambridge*, (degree, *Bachelor*)), the transformed KG with reification is the topology of mediator entity $b_k$ connecting *Alan Turing, Cambridge, Bachelor* via person, university, degree, respectively. However, without the semantic difference edge between *Alan Turing* and *Cambridge*, the fact may be wrongly reconstructed as (*Alan Turing*, obtain degree, *Bachelor*, (university, *Cambridge*)). Conversely, with semantic difference edge considered in equivalent transformation, the primary triplet can be identified with the motif structure, as depicted in Algorithm 3.
> > >
> > > **(4) Theorem 1 is trivial and not needed.**
> > >
> > > Theorem 1 guarantees the zero information loss in the transformation, which corresponds to the important information loss problem in hyperedge expansion as well as ablation study in Table 8. Based on Theorem 1 and its proof in Section C.3, we hope to make the proposed equivalent transformation model clear to the readers, as well as identify the difference between our proposed transformation with others.
> > >
> > > **(5) Theorem 2 lacks a rigid definition.**
> > >
> > > Theorem 2 Full Expressivity is a typical property for both KG and HKG modeling studies, as widely investigated in related studies (Abboud et al., 2020; Fatemi et al., 2020; Liu et al., 2021; Balazevic et al., 2019). The full expressivity guarantees the learning capacity of the HKG modeling model.
> > >
> > > Its definition is included in Section 3.2 *A HKG modeling model is fully expressive if, for any given HKG, the model can separate valid hyper-relational facts from invalid ones by appropriate parameter configuration*.
> > >
> > > **(6) About reproducabilty.**
> > >
> > > We have provided detailed implementations, code as well as data in submission for reproducibility, which is also supported by other reviewers' comments.

---

> ### Comment · Reviewer_94ho · 2022-11-22
> **Author feedback**
>
> I'd like to thank the authors for their feedback on this review.
>
> 1 Novelty. The authors still not acknowledge that the transformation is simply reification + keeping the original fact. I agree with the authors, however, that using a shared embedding is useful (whether or not its straightforward) and that an experimental validation is important.
>
> 2 Experimental study. The study uses the evaluation protocol from prior work, but it does not make a convincing case for this protocol. The ablation study I asked for is indeed present in the appendix, it should be moved to the main part. Also, it should be ensured that a separate hyperparameter search is done for this part and that entity embedding sharing is also used. Finally, with exploring the "gap", I mean: why is the proposed model better / for which kind of tasks or queries? The paper currently show s overall numbers, but does not perform deeper analysis.
>
> 3 Semantic difference unclear. First, the hyperfacts are reconstructible from reification, as the authors acknowledge. What's not reconstructible is which of the relations in the fact is "the primary one". First, this argument is only valid if a relation can act as primary in some facts and as auxiliary in others. That's neither spelled our nor does it seem to be the case. Second, it's not clear why this would matter at all for the particular task used in the experimental study. More justification/arguments are needed here.
>
> W.r.t. to Th. 1 and Th. 2, my feedback is unchanged. Theorem 1 is trivial and immediate (see 3 above). Theorem 2 lacks rigidness.
>
> I think the paper could be much stronger if it (1) downtoned its language on technical novelty and made the discussion more concise, and (2) perform a more in-depth experimental study.

---

> > ### Author Response · Authors · 2022-11-23
> > **Detailed Response to Reviewer 94h0 (Part 1)**
> >
> > We thank the reviewer for the additional discussion and constructive comments. As suggested by the reviewer, we have conducted additional experiments to explore the "gap"; these experiments primarily address the question of why is the proposed model better / for which kind of tasks or queries. Moreover, we have answered the newly raised questions as follows. We hope we have addressed the concerns and questions raised by the reviewer in a satisfactory and elucidatory manner, and look forward to further engagement with the reviewer.
> >
> > **(a-1) Novelty on the transformation.**
> >
> > Thanks for your question, and we would like to further identify the relationship between RDF reification and our proposed equivalent transformation.
> >
> > Take the sample $(s,r,o,\lbrace(a_i,v_i)\rbrace^n_{i=1})$ as an example.
> >
> > In the standard RDF reification [1, 2], the relation $r$ also becomes a node connected to the blank node (mediator entity in our case) via a newly-defined relation in the reified graph. In comparison, in the equivalent transformation, the relation $r$ connects $s,o$ with a Motif structure formed for semantics. Hence, the equivalent transformation is not simply reification + keeping the original fact, because we never transform relation $r$ into a node, which is different from RDF reification by definition.
> >
> > Despite this, we acknowledge that RDF reification is a strong motivation to our work, which will be further emphasized in the revision. In some way, our proposed equivalent transformation is an extension/modification of RDF for HKG modeling, and the reviewer's suggestion of toning down the technical novelty is constructive, which we will consider in the revision. To be specific, we will motivate this paper as a simple yet effective framework for HKG modeling inspired by RDF reification, structural and semantic modeling, but not a novel method way. Moreover, the experiments will be discussed in depth to validate the simplicity and effectiveness, and further enlighten the related research.
> >
> > Besides, thanks for your support in sharing embedding design and our experimental validation in Section 4.4.
> >
> > [1] Frey, Johannes, Kay Müller, Sebastian Hellmann, Erhard Rahm, and Maria-Esther Vidal. "Evaluation of metadata representations in RDF stores." *Semantic Web* 10, no. 2 (2019): 205-229.
> >
> > [2] Alivanistos, Dimitrios, Max Berrendorf, Michael Cochez, and Mikhail Galkin. "Query Embedding on Hyper-relational Knowledge Graphs." *arXiv preprint arXiv:2106.08166* (2021).
> >
> >
> > **(a-3) About the semantic difference.**
> >
> > First, you are right that the reconstructable argument is valid when a relation acts as primary in facts.
> >
> > Second, the reason for why such design can improve the performance is that, the semantic difference indeed exists in most datasets used for HKG evaluation.
> >
> > For example, in WikiPeople, most qualifiers in hyper-relational facts are time or space-based information (47% of hyper-relational facts in train set involve time-based information), which are the auxiliary information for the primary triple. Below are two samples converted from WikiPeople:
> >
> > eg-1: (Q375427, P54, Q190943, (P580, +2003-01-01T00:00:00Z#0#0#0#9#http://www.wikidata.org/entity/Q1985727), (P582, +2004-01-01T00:00:00Z#0#0#0#9#http://www.wikidata.org/entity/Q1985727)), interpreted as,
> >
> > (Cristiano Pereira de Souza, member of sports team, Wisła Kraków, (start time, 2003-01-01), (end time, 2004-01-01));
> >
> > eg-2: (Q372939, P19, Q19660, (P17, Q218)), interpreted as,
> >
> > (Bogdan Stelea, place of birth, Bucharest, (country, Romania)).
> >
> > Therefore, the specific design in TransEQ can capture such difference in the evaluated datasets.

---

> > > ### Author Response · Authors · 2022-11-23
> > > **Detailed Response to Reviewer 94h0 (Part 2)**
> > >
> > >
> > > **(a-2.1) About the position and setting of ablation study.**
> > >
> > > Thanks for the suggestion, and we will move it to the main part in the revision. Besides, the reported results for this part follow the same hyperparameter search as well embedding sharing adopted in the main results of Table 2.
> > >
> > > **(a-2.2) Experimental study of deeper analysis.**
> > >
> > > Thanks for the clarification on exploring "gap". According to our further analysis, the performance improvements mainly owe to the encoder introduced in TransEQ.
> > >
> > > Firstly, as tried in (Fatemi et al., 2020), the model with RDF-like conversion as well as scoring function (decoder) didn't obtain competitive performance. In comparison, our proposed TransEQ develops an encoder-decoder framework, which significantly improves the performance of original decoder models. Specifically, based on ablation study in Table 3 and Table 9, the GNN-based encoder in TransEQ brings a significant performance gain. For example, with the same decoder, CompGCN+m-DistMult (TransEQ results in Table 2) achieves an MRR of 0.569 on JF17K, while m-DistMult (No Encoder) only obtains 0.452. Similar performance improvement can be observed for the m-TransH, Transformer and HypE.
> > >
> > > Moreover, to investigate the performance for hyper-relational facts with different numbers of qualifiers, we report the breakdown performance (MRR) across groups with different numbers of qualifiers in below tables. The numbers in brackets are the ratios of testing samples with certain number of qualifiers ($n$ in $(s,r,o,\lbrace(a_i,v_i)\rbrace^n_{i=1})$) compared with the whole test set, and the results of beyond three qualifiers are omitted due to few and unreliable samples.
> > >
> > > |     |               | **WikiPeople** |              |              |
> > > | :--------------: | :-----------: | :------------: | :----------: | :----------: |
> > > | **# Qualifiers** | **0 (88.5%)** |  **1 (6.9%)**  | **2 (3.8%)** | **3 (0.6%)** |
> > > |      HINGE       |     0.35      |     0.239      |    0.289     |    0.277     |
> > > |     NeuInfer     |     0.353     |     0.322      |    0.321     |    0.251     |
> > > |       HypE       |     0.298     |      0.28      |    0.258     |    0.301     |
> > > |       RAM        |     0.407     |      0.26      |     0.26     |    0.296     |
> > > |   **TransEQ**    |   **0.477**   |   **0.343**    |  **0.344**   |   **0.41**   |
> > >
> > > |                  |               |   **JF17K**   |               |              |
> > > | :--------------: | :-----------: | :-----------: | :-----------: | :----------: |
> > > | **# Qualifiers** | **0 (42.4%)** | **1 (43.7%)** | **2 (10.5%)** | **3 (3.4%)** |
> > > |      HINGE       |     0.254     |     0.498     |     0.729     |    0.538     |
> > > |     NeuInfer     |     0.24      |     0.484     |     0.685     |    0.686     |
> > > |       HypE       |     0.308     |     0.542     |     0.711     |    0.722     |
> > > |       RAM        |     0.335     |     0.576     |     0.73      |    0.804     |
> > > |   **TransEQ**    |   **0.354**   |   **0.602**   |   **0.792**   |  **0.843**   |
> > >
> > > |                  |               | **FB-AUTO**  |               |
> > > | :--------------: | :-----------: | :----------: | :-----------: |
> > > | **# Qualifiers** | **0 (35.0%)** | **2 (2.0%)** | **3 (62.9%)** |
> > > |      HINGE       |     0.212     |    0.319     |     0.791     |
> > > |     NeuInfer     |     0.136     |    0.236     |     0.883     |
> > > |       HypE       |     0.409     |    0.325     |     0.903     |
> > > |       RAM        |   **0.549**   |    0.404     |     0.904     |
> > > |   **TransEQ**    |     0.535     |  **0.471**   |   **0.956**   |
> > >
> > > According to the results, our proposed TransEQ model achieves the best performance across most cases and datasets, i.e., various kinds of query tasks. Especially, the overall performance is highly correlated with the performance on the query case with the most samples, e.g., $n=0$ in WikiPeople and $n=3$ in FB-AUTO. Such results demonstrate that TransEQ is robust to hyper-relational facts with different numbers of qualifiers.

---

### Official Review · Reviewer_VWAf · 2022-10-25

**Confidence:** 4
**Correctness:** 2
**Technical Novelty And Significance:** 2
**Empirical Novelty And Significance:** 3
**Recommendation:** 5

**Clarity, Quality, Novelty And Reproducibility:**

I do have a few questions that may need clarification from the authors
- The description of the learnable transformations is quite confusing. How can it be performed? Any empirical/initial results?
- Are there any meaningful interpretations of the generated mediator entities and edges?

Apart from that, many details w.r.t. the model architecture and experiment implementation are discussed in the paper, which is helpful for understanding the whole paper. Code is also available.

In terms of novelty, the reviewer did not see much novelty in the model design as most of the components are taken from existing work. The most novel contribution is the HKG to KG transformation method.


**Strength And Weaknesses:**

### Strength
- The transformation idea is straightforward and well-motivated. The authors provide brief proof of the full expressivity yet it can be done in a more detailed way.
- They conducted thorough model analyses (complex analysis, comparison of model properties, hyperparameters, etc.)

### Weaknesses
- The expressiveness of the hyper-relational knowledge graph transformation is highly dependent on the choice of scoring function, which may limit the generalization capability.
- The transformation will produce a large number of additional edges and entities, making the learning and inference more expensive, as pointed out in section 3.2, they have to use parallel implementation to accelerate the process.
-  Some baselines are missing; please refer to [a][b][c[. Regarding the baselines, The reviewer is confused about the performance of StarE; according to Table 2 in the StarE paper, the best performance of StarE is higher than the reported one in this paper. For example, the best performance on WikiPeople is (MRR: 0.491; H@1:0.415, H@10: 0.648); The best on JF17K are (MRR: 0.574; H@1:0.496, H@10: 0.725). It is noted that StarE outperforms the proposed model on all metrics.


Reference:
[a] Generalizing Tensor Decomposition for N-ary Relational Knowledge Bases, WWW 2020
[b] Role-Aware Modeling for N-ary Relational Knowledge Bases, WWW 2021
[c] Searching to Sparsify Tensor Decomposition for N-ary Relational Data, WWW 2021

**Summary Of The Paper:**

In this paper, the authors proposed a model named TransEQ for hyper-relational knowledge graph modeling. In the proposed pipeline, they first convert the hyper-relational knowledge graph into normal knowledge graphs by transforming hyper-relations into mediator entities and relations while keeping the structural and semantic information; then, a graph neural network based model is used for knowledge graph encoding, and hyper-relational knowledge graph scoring functions are used for prediction.  Experiments on three datasets demonstrate the effectiveness of TransEQ.


**Summary Of The Review:**

To summarize, the paper is well-written and easy to follow. The HKG to KG transformation method is intuitive and easy to implement. However, the weaknesses of the paper cannot be ignored. The reviewer is concerned about the complexity, generalization capability, and correctness of the evaluation, which makes the reviewer give a negative recommendation.

---

> ### Author Response · Authors · 2022-11-19
> **Detailed Response to Reviewer VWAf (Part 1)**
>
>
> We want to thank the reviewer for their valuable questions and suggestions and in particular their significant time investment. We will address your concerns in the following.
>
> **(1) The model expressiveness is highly dependent on scoring function choice.**
>
> Yes, you are right that the expressiveness of our proposed TransEQ model depends on the choice of scoring function in the decoder part.
>
> However, it is precise because of the flexible encoder-decoder design that TransEQ can generalize to more datasets and scenarios.
>
> Especially, any scoring function defined in existing HKG modeling studies and hopefully future studies can be chosen as the decoder part in TransEQ, In other words, the generalization capability of TransEQ is at least as equal as the HKG model adopted in decoder part.
>
> To validate the generalization capability, we conduct additional experiments to compare TransEQ with fully expressive model BoxE (Abboud et al., 2020) as decoder, denoted as TransEQ (BoxE), as shown in below Table. The originally reported TransEQ with m-DistMult model as decoder in Table 2 is denoted as TransEQ (m-DistMult) here:
> | Model                | MRR   | Hit@1 | Hit@10 |
> |----------------------|-------|-------|--------|
> | G-MPNN               | 0.763 | 0.724 | 0.838  |
> | BoxE                 | 0.844 | 0.814 | 0.898  |
> | TransEQ (m-DistMult) | 0.870 | 0.842 | 0.909  |
> | TransEQ (BoxE)       | 0.880 | 0.859 | 0.920  |
>
> The results further confirm the better performance of TransEQ with more expressive scoring function of BoxE. Thus, we show the flexible encoder-decoder design as a strength for model generalization.
>
>
> **(2) About the learning and inference cost of the transformation.**
>
> We want to clarify that the parallel implementation only means the training and inference by GPU, but not extra engineering techniques for acceleration, which can be checked in our provided codes.
>
> Especially, Figure 7 in Section D.3 presents the comparison of training time of different models. All these models are run on a single GPU without extra parallel design, and we can observe that TransEQ is much more efficient than other baselines. The same phenomenon can also be found in inference step.
>
> Therefore, the implementation of TransEQ is simple and effective, which can be applied for practical use.
>
> **(3) About the missing baselines.**
>
> We didn't miss the baselines raised by the reviewer.
>
> Reference [b] and [c] correspond to RAM (Liu et al., 2021) and S2S (Di et al., 2021) in Table 2.
>
> Reference [a] corresponds to GETD (Liu et al., 2020) in Table 11.
>
> Our proposed TransEQ outperforms such baselines as shown in the paper.
>
> **(4) About the performance of StarE with their reported ones in paper.**
>
> The reported results of StarE in our paper are right, since we adopt a more practical evaluation setting compared with the original setting used in StarE paper, as we mentioned in **Task and Evaluation Metrics** part in Section 4.1. The reported results of StarE in our paper are copied from (Di & Chen, 2022), which follow the same setting and dataset splits as ours.
>
> Given a testing sample $x:=(s,r,o,{\lbrace(a_i,v_i)\rbrace}^n_{i=1})$, the original setting in StarE assumes that only $s$ and $o$ may be missing, which is actually impractical because the value entities $v1,\cdots,v_n$ may also be missing. Thus, we follow the setting in (Di & Chen, 2022; Liu et al., 2021; Fatemi et al., 2020; Abboud et al., 2020):
>
> For the testing sample $x:=(s,r,o,{\lbrace(a_i,v_i)\rbrace}^n_{i=1})$, we assume that either entity in $s, o, v_1,\cdots,v_n$ may be missing, and apply the model to predict missing entity at each position in evaluation. Especially, we first divide testing samples into groups based on the number of qualifiers, and test for each group and report the average performance of all samples. For example, we calculate twice for $(s,r,o)$ in binary group, three times for $(s,r,o,\lbrace(a_1,v_1)\rbrace)$ in ternary group and four times for $(s,r,o,\lbrace(a_1,v_1), (a_2,v_2)\rbrace)$ in quaternary group, and all these results are averaged together for the reported performance.

---

> > ### Author Response · Authors · 2022-11-21
> > **Detailed Response to Reviewer VWAf (Part 2)**
> >
> >
> > **(5) About the learnable transformation.**
> >
> > The learnable transformation can be identified as an adjacent matrix learning or graph structure learning problem.
> >
> > Take the illustration in Figure 3 as an example.
> >
> > Given a hyper-relational fact with involved entities $s, o, v_1, v_2$ and mediator entity $b_k$, the proposed equivalent transformation is equivalent to an adjacent matrix (The columns and rows correspond to $s, o, v_1, v_2, b_k$):
> > $$ A=\left[
> > \begin{matrix}
> >  0 & 1 & 0 & 0 & 0 ;\\
> >   0 & 0 & 0 & 0 & 0 ;\\
> >  0 & 0 & 0 & 0 & 0 ;\\
> >  0 & 0 & 0 & 0 & 0; \\
> >  1 & 1 & 1 & 1 & 0
> > \end{matrix}
> >   \right]
> > $$
> > Such matrix is fixed due to our manual design of equivalent transformation. The learnable transformation is to make such matrix learnable in the modeling. For example, we can train a neural network model with entities and relations as input, and output an adjacent matrix to replace the manually designed transformation here, i.e., $A=f(s, o, v_1, v_2, b_k, r, r_1, r_2)$. The matrix learning can be combined with HKG modeling for joint training. Possible references are included below.
> >
> > However, as we stated in **Manually-designed v.s. Learnable Transformations** part, the learning process is over complex without theoretical guarantee, while TransEQ with the manually-designed transformation has achieved state-of-the-art performance.
> >
> > Thus, the learnable transformation can be future work.
> >
> > \[1\] Yu, Yue, Jie Chen, Tian Gao, and Mo Yu. "DAG-GNN: DAG structure learning with graph neural networks." In ICML, 2019.
> >
> > \[2\] Zheng, Xun, Bryon Aragam, Pradeep K. Ravikumar, and Eric P. Xing. "Dags with oo tears: Continuous optimization for structure learning." NeurIPS, 2018.
> >
> >
> > **(6) About the meaningful interpretations of mediator entities and edges.**
> >
> > Thanks for the suggestion. For now we mainly visualize the learnt embeddings of mediator entities to reveal the semantics, as shown in Figure 4(a). The clustering results therein can interpret the semantic relatedness in HKG: hyper-relational facts with the same primary relation cluster together. We will try to identify more meaningful results in future work.

---

### Official Review · Reviewer_cp9m · 2022-11-03

**Confidence:** 3
**Correctness:** 3
**Technical Novelty And Significance:** 2
**Empirical Novelty And Significance:** 2
**Recommendation:** 5

**Clarity, Quality, Novelty And Reproducibility:**

The paper presentation is unclear in which more information is needed to judge the work.

Even though the method of creating a graph transformation is widely used in other domain, there is still novelty in this work to introduce it into HKG modeling.

The source code is released which the reproducibility can be easily tested.


**Strength And Weaknesses:**

Strength:
1. The proposed method is intuitive and effective
2. The proposed graph transformation bridges the algorithms between KG modeling and HKG modeling.

Weaknesses:
1. In Sec 3.1.2 under the GNN-based Encoder. Initialized embedding alpha setting is kind of ad-hoc (grid search in the experiment). An insightful guidance for setting such hyper-params for new dataset will be helpful.
2. In Model Training part. The negative sampling procedure is unclear since the graph itself is not the graph original HKG task. The negative sampling procedure may influence the result.
3. Possibly unfair comparison in the experiment. In Sec 4.2, the author indicated the task in their setting is different from StarE, which include more predicting task. However, important detailed information is missing such as how they handle facts with different number of qualifiers, how they adapt the baseline which doesn't support predicting entities in qualifiers into their setting.
4. A lot of explanations are in appendix, which block the reader from following the main stream of the paper.

**Summary Of The Paper:**

This paper presents a comprehensive method to model the hyper-relational knowledge graph. The authors transform a HKG into a KG by introducing extra entities and relations which constructing a one-to-one mapping between the constructed KG and the previous HKG. The transformation is information preserving which is intuitive and widely used in other research domain such as graph theory. The author then design cured  function to aggregate information for newly created entities. And feed into an Encoder-Decoder framework for model training. The experiment result for three datasets shows the improvement for such graph transformation.

**Summary Of The Review:**

The proposed method is simple yet effective. The technical contribution is limited. The experiment settings omit a lot of information which prevent fair comparison. Hence I would recommend "Discussion" to get feedback from the authors.

---

> ### Author Response · Authors · 2022-11-19
> **Detailed Response to Reviewer cp9m**
>
> We want to thank the reviewer for their constructive and helpful feedback, and in particular their significant time investment. To ease the identification of additions to our paper, we have marked added sections and texts in response to your questions in green. We will address your concerns in the following.

---

> > ### Author Response · Authors · 2022-11-21
> > **Detailed Response to Reviewer cp9m (Part 1)**
> >
> >
> > **(1) An insightful guidance for setting the hyper-parameter $\alpha$.**
> >
> > Thanks for the suggestion. The experiments should set a large value for hyper-parameter $\alpha$ on large datasets, and a small one on small datasets.
> >
> > To be specific, $\alpha$ is used to tune the sharing embedding ratio for mediator entities. According to the parameter complexity with mediator entities, i.e., $\mathcal{O}(N\cdot(1-\alpha)\cdot d+n^{\text{pri}}_r\cdot\alpha\cdot d)$ with $N$, $n^{\text{pri}}_r$ and $d$ as the number mediator entities, the number of primary relations and embedding dimension, respectively ($N\gg n^{\text{pri}}_r$), a smaller $\alpha$ brings more embedding parameters. Hence, the selection of $\alpha$ can be viewed as a bias-variance tradeoff for the model. A small $\alpha$ leads to a complex model, which tends to have a lower bias but a higher variance for overfitting, while a large $\alpha$ leads to a simple model, which tends to have a higher bias but a lower variance for underfitting.
> >
> > According to our empirical experiments, datasets with many training facts and relations (like WikiPeople and JF17K) should select a large value for $\alpha$ value to avoid tremendous model parameters and overfitting, while datasets with limited facts and relations (like FB-AUTO) can adopt a small value for $\alpha$ to avoid underfitting. The below table summarizes the related statistics and optimal $\alpha$ value on datasets, which is in accord with our guidance.
> > |    Dataset    |     $\vert\mathcal{R}\vert$     | #Train | $\alpha$ |
> > |:----------:|:-----------------------:|:-----------------------:|:--------:|
> > |   FB-AUTO  |            8            |        6,778          |    0.2   |
> > |    JF17K   |           322           |            61,104       |    0.6   |
> > | WikiPeople |           707           |          305,725        |    0.8   |
> >
> > **(2) Will the negative sampling influence the result.**
> >
> > The negative sampling procedure will not influence the result, and such negative sampling as well as loss functions are commonly used in HKG modeling (Fatemi et al., 2020; Liu et al., 2020; Abboud et al., 2020).
> >
> > Take predicting the missing object entity for example. The overall training procedure is as follows:
> >
> > * Convert the HKG into KG with equivalent transformation;
> > * Apply GNN-based encoder to learn representations of entities and relations;
> > * For each training sample $x:=(s,r,o,{\lbrace(a_i,v_i)\rbrace}^n_{i=1})\in\mathcal{F}^H$, construct neagtive samples $x^\prime:=(s,r,o^\prime,{\lbrace(a_i,v_i)\rbrace}^n_{i=1})\in\mathcal{F}^H$ with all $o^\prime\in\mathcal{E}$, i.e., we sample $\vert\mathcal{E}\vert-1$ negative samples;
> > * Use the representations output from encoder and calculate the loss with the scores of $\phi(x),\phi(x^\prime)$;
> > * Update model parameters by gradient backward algorithm.
> >
> > Therefore, we model all entities except the predicting one as negative samples, and the negative sampling is directly executed by replacing the object entity instead of sampling on graphs.
> > The overall training procedure is provided in Section B.2 for better understanding.
> >
> >
> > **(3) About the unfair comparison in the experiment.**
> >
> > The comparisons with all baselines in experiments are fair. We directly copy the results of baselines from (Di & Chen, 2022; Liu et al., 2021; Fatemi et al., 2020;) or their original papers, which follow the same setting and dataset splits as ours. The StarE results are copied from (Di & Chen, 2022)
> >
> > Given a testing sample $x:=(s,r,o,{\lbrace(a_i,v_i)\rbrace}^n_{i=1})$, the original setting in StarE assumes that only $s$ and $o$ may be missing, which is actually impractical because the value entities $v1,\cdots,v_n$ may also be missing. Thus, we follow the setting in (Di & Chen, 2022; Liu et al., 2021; Fatemi et al., 2020; Abboud et al., 2020):
> >
> > For the testing sample $x:=(s,r,o,{\lbrace(a_i,v_i)\rbrace}^n_{i=1})$, we assume that either entity in $s, o, v_1,\cdots,v_n$ may be missing, and apply the model to predict missing entity at each position in evaluation. Especially, we first divide testing samples into groups based on the number of qualifiers, and test for each group and report the average performance of all samples. For example, we calculate twice for $(s,r,o)$ in binary group, three times for $(s,r,o,\lbrace(a_1,v_1)\rbrace)$ in ternary group and four times for $(s,r,o,\lbrace(a_1,v_1), (a_2,v_2)\rbrace)$ in quaternary group, and all these results are averaged together for the reported performance.
> >
> > As for baselines like StarE, they originally supported our setting of predicting entities in qualifiers but didn't adopt in their paper. Thus, we can easily modify their evaluation code for our more generalized setting.
> >
> > **(4) The structure of the paper.**
> >
> > Thanks for the suggestion. We have revised the paper to keep a clear mainstream for reading and understanding.

---

### Decision · Program_Chairs · 2023-01-20

**Decision:**

Reject

**Justification For Why Not Higher Score:**

See above in the meta-review for why this paper was not recommended for acceptance.

**Justification For Why Not Lower Score:**

N/A.

**Metareview: Summary, Strengths And Weaknesses:**

The authors propose a simple transformation approach to hyper-relational knowledge graph (HKG) modeling with semantic and structural information in this paper. All reviewers agree that the HKG direction is important and a good way to enrich the standard triple-based knowledge graph data structure. However, only one reviewer voted to accept this paper, and three vetoed it. The primary reasons that the reviewers consider this paper not ready for ICLR 2023 is that: 1. The experiments could be strengthened. 2. The choice of scoring function highly affects the generalization. 3. The transformation might make the learning and inference intractable. 4. There are concerns about novelty and insights into the results. Overall, the panel has decided not to recommend this paper for acceptance at ICLR at this time.

**Summary Of Ac-Reviewer Meeting:**

No AC-reviewer meeting was scheduled because it was not a borderline paper.